# Prediction Models That Learn to Avoid Missing Values

**Lena Stempfle** [1] [*]   **Anton Matsson** [1] [*]   **Newton Mwai** [1]   **Fredrik D. Johansson** [1]

## Abstract

Handling missing values at test time is challenging for machine learning models, especially when aiming for both high accuracy and interpretability. Established approaches often add bias through imputation or excessive model complexity via missingness indicators. Moreover, either method can obscure interpretability, making it harder to understand how the model utilizes the observed variables in predictions. We propose *missingness-avoiding* (MA) machine learning, a general framework for training models to rarely require the values of missing (or imputed) features at test time. We create tailored MA learning algorithms for decision trees, tree ensembles, and sparse linear models by incorporating classifier-specific regularization terms in their learning objectives. The tree-based models leverage contextual missingness by reducing reliance on missing values based on the observed context. Experiments on real-world datasets demonstrate that `MA-DT`, `MA-LASSO`, `MA-RF`, and `MA-GBT` effectively reduce the reliance on features with missing values while maintaining predictive performance competitive with their unregularized counterparts. This shows that our framework gives practitioners a powerful tool to maintain interpretability in predictions with test-time missing values.

## 1. Introduction

Missing values complicate the deployment of prediction models. In applications where humans provide inputs to models and read off their output, such as when using clinical risk scores, users must either assign values to unobserved variables or rely on the model to handle this.

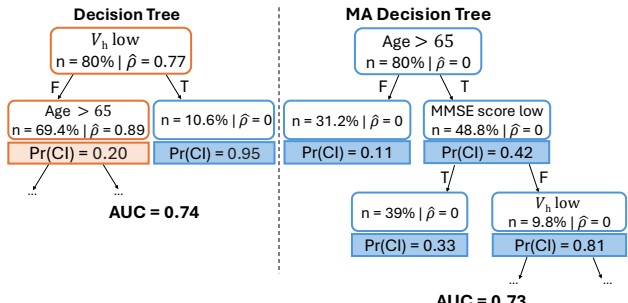

*Figure 1.* Two decision trees built to diagnose cognitive impairment (CI) in adult patients. The left side illustrates a regular decision tree fit solely for accuracy, while the right shows a missingness-avoiding (MA) tree, which incorporates a regularization parameter $\alpha$ to reduce reliance on missing values. The regular tree initially splits on a positive MRI scan outcome, determined by hippocampal volume ($V_h$). However, this feature is missing for many patients who did not undergo a scan, resulting in high reliance on missing values (indicated by a large $\rho$). In contrast, the MA tree achieves a comparable AUC while avoiding reliance on missing data entirely ($\rho = 0$). Orange nodes indicate missing values along the decision path, while blue denotes no missingness reliance. Note that zero imputation is used, resulting in complete observations in the right branch of the regular tree.

This can introduce bias and undermine the model's interpretability, which may be crucial in safety-critical fields like healthcare.

Classically, research on missing values has focused on learning from incomplete data, fitting regression models to imputed values, and applying these to complete cases (Rubin, 1976). However, when missing values also occur in deployment (at "test time"), the impute-then-regress strategy is needlessly restrictive. When the distribution of observed and missing values remains unchanged, Bayes-optimal predictions can be made without imputation, based on observed values and a missingness mask—indicators for which values are missing. Attempts to achieve the same result using high-capacity imputation models can increase bias (Le Morvan et al., 2021), fail to use informative missingness (Rubin, 1976), and reduce the interpretability of predictions.

Fitting flexible prediction models to trivially imputed inputs (e.g., zero imputation) and the missingness mask

*Equal contribution  [1]Department of Computer Science and Engineering, Chalmers University of Technology and University of Gothenburg, SE-41296 Gothenburg, Sweden. Correspondence to: Lena Stempfle <stempfle@chalmers.se>.

*Proceedings of the $42^{nd}$ International Conference on Machine Learning*, Vancouver, Canada. PMLR 267, 2025. Copyright 2025 by the author(s).

is well justified when accuracy is the primary target. Such models often achieve strong predictive performance (Van Ness et al., 2023) but may inadvertently rely on complex dependencies between the missingness mask, observed features, and imputed values, making their decision process challenging to interpret (Stempfle et al., 2023; van Smeden et al., 2020).

A small class of methods avoids both imputation and prediction with indicators by handling missing values natively. For example, XGBoost (Chen & Guestrin, 2016) and other tree-based models can learn "default" decision paths for missing values (Chen & Guestrin, 2016; Pedregosa et al., 2011). Probabilistic models, like Bayesian networks (Pearl, 2014), or variational autoencoders (Kingma & Welling, 2013) can marginalize over missing variables rather than imputing them (Ma & Chen, 2018). Currently, however, native handling of missing values is limited to specialized models.

What if we could sidestep imputation, indicators, and specialized architectures by *learning to avoid* using features with missing values in prediction? If a feature is missing but not useful for prediction for a particular input, a good model should not need access to it. Recently, Stempfle & Johansson (2024) made use of a related idea by training disjunctive generalized linear rule models (GRLMs) to exploit covariate redundancy in datasets and learn rules whose values can be evaluated even if some variables in the input are missing. The approach is promising, but limited to GRLMs and does not extend to ensemble models, which better capture complex, nonlinear interactions.

Figure 1 illustrates a scenario where a standard decision tree (left) achieves high accuracy ($\mathrm{AUC} = 0.74$) in predicting cognitive impairment (CI) but relies on frequently missing features, such as MRI results (hippocampal volume $V_h$), that are unavailable for $77\%$ of patients (indicated by $\hat{\rho} = 0.77$). Patients without MRI results most likely showed no signs of CI and could be classified well based on a cognitive test score alone, such as the Mini–Mental State Examination (MMSE) score. The tree on the right achieves comparable predictive performance ($\mathrm{AUC} = 0.73$) without ever asking for a feature that is missing ($\hat{\rho} = 0$): MRI results are only used for patients with low MMSE scores. This work proposes a method for learning such trees by strategically minimizing reliance on missing values, and exploiting informative structure in the data. While avoiding missing values at test time completely may not be feasible, our approach reduces dependence on missing features while maintaining predictive performance.

**Contributions.** We introduce *missingness-avoiding* (MA) machine learning, reconciling accuracy with minimal reliance on features with missing values at test

time. We (1) implement this idea for decision trees (`MA-DT`), sparse linear models (`MA-LASSO`), and ensemble methods (`MA-RF` and `MA-GBT`), providing fit-predict implementations for seamless integration; (2) analyze when MA learning is feasible without sacrificing predictive performance; and (3) demonstrate that MA models perform comparably to baselines with low reliance on missing values across real-world datasets: confidence intervals for the AUROC of `MA-LASSO`, `MA-DT`, and `MA-RF` and their unregularized counterparts overlap in nearly all cases, while the MA models consistently and substantially reduce missingness reliance, in some cases from $100\%$ to $0\%$.

## 2. Related Work

As described in the introduction, *impute-then-regress* procedures, tree-based methods, and *missingness indicators* are common strategies for handling missing values in supervised learning. While *missingness indicators* can reduce prediction errors when missingness is informative (Van Ness et al., 2023), they may also introduce spurious correlations (Kyono et al., 2021), feedback loops (van Smeden et al., 2020), and reduced interpretability by effectively doubling the feature set. To address these challenges, McTavish et al. (2024) proposed M-GAM, a generalized additive model that learns sparse interactions between features and the missingness mask. Although this approach improves interpretability, it does not explicitly restrict reliance on missing values—a key objective of our work. Additionally, neural architectures like NeuMiss (Le Morvan et al., 2020) integrate missing patterns directly into model training, thereby avoiding imputation. However, this comes at the cost of reduced interpretability.

Native handling of missing values in tree-based models can be achieved through "Missingness Incorporated in Attributes" (MIA), which treats missing values as a separate category (Twala et al., 2008; Kapelner & Bleich, 2015; Josse et al., 2024). While effective, this approach is typically restricted to a single model class and is incomparable across models. Some studies instead leverage structured missingness. For example, Fletcher Mercaldo & Blume (2020) proposed fitting separate models per missing pattern, so-called pattern submodels, but this can lead to inefficient use of data. Stempfle et al. (2023) addressed this limitation by introducing sparsity-induced parameter sharing, which improves generalization and consistency while preserving interpretability. However, these approaches are limited to linear models and do not extend easily to more complex settings.

Branch-exclusive splits trees (BEST), proposed by Beaulac & Rosenthal (2020), allow users to guide the learning algorithm during the data partitioning process by specifying

gating variables—features that restrict the use of other variables for splitting outside designated regions of the predictor space. BEST is particularly effective when there is strong prior knowledge about the data-generating process, such as in the ODDC rule setting described in Section 5.1. However, its performance may suffer when the gating variables have low predictive value, and the use of explicit mask splits can reduce interpretability by causing the tree structure to be dominated by missingness logic.

## 3. Problem Statement

Let $X = [X_1, ..., X_d]^\top \in \mathcal{X}$ be the input to a hypothesis $h \in \mathcal{H}$, used to predict an outcome $Y \in \mathcal{Y}$. The input consists of $d$ tabular (numerical or categorical) variables, which may be missing in some observations, taking the value **na**. Hence, $\mathcal{X} \subseteq (\mathbb{R} \cup \{\mathbf{na}\})^d$. We view $X$ as generated by applying a (random) missingness mask $M \in \{0, 1\}^d$ to a complete input variable $X^c$. For each variable $j \in [d]$,

$$X_j = \begin{cases} X_j^c, & \text{if } M_j = 0 \\ \mathbf{na}, & \text{if } M_j = 1 \end{cases} .$$

We assume that the observed variables follow a *fixed, unknown* distribution $p(X, M, Y)$, which remains the same during training and test time. Algorithms learn from a dataset $\mathcal{D} = \{(\boldsymbol{x}_i, \boldsymbol{m}_i, y_i)\}_{i=1}^n$ of $n$ samples assumed to be drawn i.i.d., $(\boldsymbol{x}_i, \boldsymbol{m}_i, y_i) \sim p(X, M, Y)$, where $\boldsymbol{x}_i = [x_{i,1}, ..., x_{i,d}]^\top$.

A common strategy to learn $h$ is to first impute the missing values in $X$, obtaining an imputed dataset $X^I = \{\boldsymbol{x}_i^I\}_{i=1}^n$ with imputed observations $\boldsymbol{x}_i^I$. The model is then trained on observed or imputed covariates—a strategy known as *impute-then-regress* estimation. If missingness is informative, $p(M \mid Y) \neq p(M)$, incorporating the missingness mask $\boldsymbol{m}$ as binary indicators into the model input can reduce prediction error, improving downstream performance (Van Ness et al., 2023; Rubin, 1976).

Our goal is to learn hypotheses $h$ that are *missingness avoiding*, that is, they are unlikely to require the value of a missing variable at test time. Let $a_h(\boldsymbol{x}, j) = 1$ denote the event that computing $h(\boldsymbol{x})$ requires access to the value of $x_j$ (imputed or observed) and $a_h(\boldsymbol{x}, j) = 0$ otherwise. For example, computing the prediction of a linear model with imputed inputs, $h_\theta(\boldsymbol{x}^I) = \theta^\top \boldsymbol{x}^I$, requires access to $x_j^I$ whenever $\theta_j \neq 0$. A decision tree requires access to $x_j^I$ if feature $j$ appears on the prediction path from root to leaf for the input $\boldsymbol{x}$. A rule model requires access to $x_j^I$ if the truth values of its rules are contingent on $x_j^I$. Figure 2 illustrates how different models may avoid relying on missing values.

Stempfle & Johansson (2024) introduced *missingness re-*

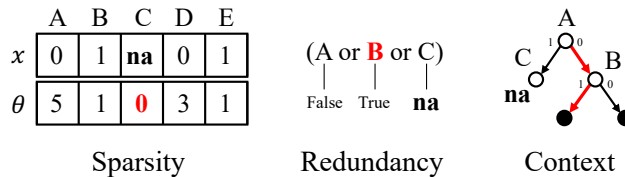

| | A | B | C | D | E |
|---|---|---|---|---|---|
| $x$ | 0 | 1 | **na** | 0 | 1 |
| $\theta$ | 5 | 1 | **0** | 3 | 1 |

Sparsity | Redundancy | Context

*Figure 2.* Missing values can be avoided in several ways. Sparse models (left) can be trained to not use features that are frequently missing. Disjunctive rule models (middle) can be fit to include rules that exploit redundancy in the variable set. Trees (right) can be fit so that missing values rarely occur on the decision paths.

*liance* with the definition (formalized here) that $h$ relies on missing values in an observation $\boldsymbol{x}$ if there is a feature $j$ such that 1) $x_j = \mathbf{na}$ and 2) computing $h(\boldsymbol{x})$ requires evaluating $x_j$ or its imputed value $x_j^I$. We use a binary indicator function $\rho(h, \boldsymbol{x}) \in \{0, 1\}$ to indicate that computing $h(\boldsymbol{x})$ relies on at least one missing feature in $\boldsymbol{x}$,

$$\rho(h, \boldsymbol{x}) = \max_{j \in [d]} \mathbb{1}[a_h(\boldsymbol{x}, j) = 1 \wedge x_j = \mathbf{na}] . \quad (1)$$

In principle, we could define missingness reliance for an instance $\boldsymbol{x}$ by aggregating over features using functions $\varphi$ other than the $\max$ operator, $\rho_\varphi(h, \boldsymbol{x}) = \varphi(\{\mathbb{1}[a_h(\boldsymbol{x}, j) = 1 \wedge x_j = \mathbf{na}]\}_{j=1}^d)$. For example, $\varphi$ could compute the average over the set, $\varphi(\cdot) = \frac{1}{d} \sum_{i=1}^d (\cdot)_i$, and the corresponding reliance $\rho_\varphi(h, \boldsymbol{x})$ would represent the fraction of missing values in $\boldsymbol{x}$ relied on by $h$. In this paper, we restrict ourselves to binary reliance based on $\max$: Does $h$ rely on missing values when predicting for $\boldsymbol{x}$ or not?

The expected missingness reliance $\rho$ of a hypothesis $h$ in a distribution $p(X, M, Y)$ is then defined as $\rho(h) := \mathbb{E}_p[\rho(h, X)]$. The goal of MA learning is to find a suitable trade-off between expected predictive performance and missingness reliance:

$$\underset{h \in \mathcal{H}}{\text{minimize}} \ \mathbb{E}_p[L(Y, h(X))] + \alpha \rho(h) , \quad (2)$$

controlled by a tradeoff parameter $\alpha \geq 0$.

## 4. Missingness-Avoiding Prediction Models

Next, we introduce *missingness-avoiding* (MA) learning algorithms for decision trees, sparse linear models, and tree ensembles, showing that classifier-specific regularization can reduce reliance on missing features across diverse model classes. Decision trees capture nonlinear interactions, sparse linear models offer interpretability, and tree ensembles boost generalization and performance.

### 4.1. Missingness-Avoiding Decision Trees

Trees are inherently well suited to avoid missing values as their reliance on input features is contextual: only a sub-

set of observations will pass through a given node on their decision path, depending on the values of other features. This property naturally extends to the binary decision tree $h$ which can be defined recursively by a root node $u_0$ and its two children $l_0, r_0$, their children, grandchildren, and so on. Each internal node $u \in \mathcal{V}$ in the tree is endowed with a decision stub $h_u : \mathcal{X} \to \{0, 1\}$ which partitions $\mathcal{X}$ into a "left" and "right" half. Here, we restrict ourselves to the typical univariate threshold stubs, $h_u(\boldsymbol{x}) = \mathbb{1}[x_{j_u} > \tau_u]$ for a feature $j_u$ and threshold $\tau_u$.

Let $\mathcal{L}$ denote the set of leaves of the tree—nodes with no children. Each leaf $\ell \in \mathcal{L}$ has a value $\hat{y}_\ell \in \mathcal{Y}$ used for prediction. The *decision path* $\pi_h(\boldsymbol{x}) = (u_0, ..., u_t)$ for an input $\boldsymbol{x}$ in a tree $h$ is the sequence of nodes traversed from $u_0$ to a leaf $u_t \in \mathcal{L}$ such that for all $k = 0, ..., t-1$,

$$u_{k+1} = \begin{cases} l_{u_k}, & x_{j_{u_k}} \leq \tau_u \\ r_{u_k}, & x_{j_{u_k}} > \tau_u \end{cases} .$$

We say that a feature $x_j$ is on the decision path for an input $\boldsymbol{x}$ if there exists a node $u \in \pi_h(\boldsymbol{x})$ such that $j_u = j$. Finally, we can define the missingness reliance for the tree in $\boldsymbol{x}$ as

$$\rho(h, \boldsymbol{x}) := \max_{u \in \pi_h(\boldsymbol{x})} \mathbb{1}[x_{j_u} = \mathbf{na}] . \quad (3)$$

That is, a tree $h$ relies on a missing value in input $\boldsymbol{x}$ if there is a node on the decision path $\pi_h(\boldsymbol{x})$ where the decision is made based on a feature $j$ which has a missing value in $\boldsymbol{x}$.

FITTING TREES WITH SMALL MISSINGNESS RELIANCE

To fit trees that minimize prediction error and reliance on missing values, we propose modifications to greedy splitting algorithms like C4.5 (Quinlan, 2014) or ID3 (Quinlan, 1986), as these dominate decision tree implementations. Global optimization approaches, such as MIP formulations (Verwer & Zhang, 2019) and other solvers (Gurobi Optimization, LLC, 2024; ILOG, IBM, 2010; De Moura & Bjørner, 2008), could also be regularized with missingness reliance.

Greedy splitting partitions a dataset $\mathcal{D}$ by sequentially splitting a leaf $\ell$ in a tree using a feature $j$ and threshold $\tau$, chosen by minimizing a splitting criterion $C(\ell, \mathcal{D}; j, \tau)$, such as Gini impurity or information gain (Hastie et al., 2009). Let $S_\ell := \{i \in [n] : \ell \in \pi(\boldsymbol{x}_i)\}$ be the index set of samples in $\mathcal{D}$ assigned to leaf $\ell$ in $h$. We regularize a given criterion $C$ to reduce reliance on missingness values, splitting leaves $\ell$ based on feature-threshold pairs $(j, \tau)$ that solve the following optimization problem, where $\alpha \geq 0$ controls the regularization strength:

$$\underset{j \in [d], \tau \in \mathbb{R}}{\text{minimize}} \ C(\ell, \mathcal{D}; j, \tau) + \alpha \sum_{i \in S_\ell} \frac{\sigma_{i,j}}{|S_\ell|} \mathbb{1}[x_{i,j} = \mathbf{na}] . \quad (4)$$

We include a sample-specific feature weight $\sigma_{i,j} \in \{0, 1\}$ to enable generalization to tree ensembles, as discussed in Section 4.3. For a single tree, we set $\sigma_{i,j} = 1$ for all $i, j$. However, in principle, we could set $\sigma_{i,j} = 0$ to remove the missingness reliance penalty when feature $j$ is already used in a split along the decision path $\pi(\boldsymbol{x}_i)$, so that further splits on the same feature would not increase the overall missingness reliance. The criterion $C$ may or may not operate on imputed data or use default rules like, for example, XGBoost (Chen & Guestrin, 2016). We refer to trees that minimize this regularized criterion as MA decision trees, or `MA-DT` for short.

### 4.2. Missingness-Avoiding Sparse Linear Models

Our framework extends naturally to sparse linear models. For a linear model $h(\boldsymbol{x}) = \theta^\top \boldsymbol{x}$, we define the missingness reliance as

$$\rho(h, \boldsymbol{x}) := \max_{j \in [d]} \mathbb{1}\left[|\theta_j| > 0\right] \mathbb{1}\left[x_j = \mathbf{na}\right] ,$$

with $\rho(h) := \mathbb{E}_p[\rho(h, X)]$ as stated in Section 3. As is clear from this definition, linear models *cannot* contextually avoid missing values like trees do. No matter the values of other features, every time a feature $j$ used by the model is missing, it counts toward $\rho(h)$. Nevertheless, we can reduce reliance on missing values through variable selection by penalizing coefficients $\theta_j$ of variables with a high frequency of missingness $\overline{m}_j$, and solving

$$\underset{\theta \in \mathbb{R}^d}{\text{minimize}} \ \frac{1}{n} \sum_{i=1}^{n} L(y_i, \theta^\top \boldsymbol{x}_i^I) + \sum_{j=1}^{d} (\lambda + \alpha \overline{m}_j)|\theta_j| \quad (5)$$

with parameter $\lambda > 0$ and $\alpha \geq 0$ controlling the regularization strength. Here, $\overline{m}_j = \frac{1}{n} \sum_{i=1}^{n} m_{i,j}$ is the empirical missingness rate of feature $j$. We refer to models that minimize (5) as `MA-LASSO`.

The standard Lasso (Tibshirani, 1996) encourages sparsities in linear models by applying an $L^1$-penalty proportional to the sum of absolute magnitudes of the coefficients, multiplied by a parameter $\alpha > 0$. In `MA-LASSO`, we introduce a feature-specific penalty $\lambda_j = \lambda + \alpha \overline{m}_j$, where regularization is adjusted based on the missingness frequency of each feature $j$. Features with higher missing proportions receive larger penalties, making them more likely to be dropped from the model. We can solve (5) by applying standard Lasso with parameter $\lambda'$ to a dataset with rescaled features $\tilde{x}_{i,j} = \frac{\lambda'}{\lambda_j} x_{i,j}$. This approach can be extended to generalized linear models and generalized additive models.

**Why Lasso?** While $L^1$ regularization is widely used to induce sparsity in learned models, it is primarily motivated as a convex relaxation of $L^0$ regularization, which explicitly penalizes the cardinality of the selected variable

set. Here, we use the $L^1$ penalty due to its compatibility with widely used implementations in packages like scikit-learn (Pedregosa et al., 2011). However, (5) may be analogously defined using the $L^0$ norm and, in future work, it would be interesting to explore solving the resulting problem directly (Liu et al., 2022; Dedieu et al., 2021).

### 4.3. Missingness-Avoiding Tree Ensembles

Tree ensembles, such as random forests and gradient-boosted trees, typically outperform single trees in terms of predictive accuracy (Curth et al., 2024). We can combine several MA trees into an MA tree ensemble, following the same principles used for standard CART ensembles. For an ensemble $e = (h_1, ..., h_M)$ of $M$ estimators, we generalize the definition of missingness reliance for trees applied to an input $\boldsymbol{x}$ as

$$\rho(e, \boldsymbol{x}) := \max_{h \in e} \max_{u \in \pi_h(\boldsymbol{x})} \mathbb{1}[x_{j_u} = \mathbf{na}] , \qquad (6)$$

where we recall that $\pi_h(\boldsymbol{x})$ is the decision path for $\boldsymbol{x}$ in $h$. Since adding models to an ensemble can only increase the average reliance on a variable, extending the ensemble can only increase the missingness reliance.

We implement both an MA random forest (`MA-RF`) as well as MA gradient boosted trees (`MA-GBT`). Our implementations are based on the corresponding meta-estimators in the scikit-learn library, with MA trees replacing the default tree estimators for both classification and regression tasks. In `MA-RF`, the individual trees are fit independently, allowing the tree-building process to be parallelized. The parameter $\sigma_{i,j}$ in the regularized node splitting criterion in (4) is set to 1 for all $i, j$.

`MA-GBT` is a gradient-boosting classifier specifically designed to minimize missingness reliance. At the core, it follows the structure of standard gradient-boosting, where the model $h_m(\boldsymbol{x})$ added at step $m$ fits the pseudo-residuals of the ensemble $e_{m-1}(\boldsymbol{x})$ learned up to that point using the splitting objective in (4) with a regression criterion (mean squared error) for $C$. See Algorithm 1 in Appendix B for the pseudo algorithm for `MA-GBT`. Unlike `MA-RF`, each model in the boosting process builds on the reliance patterns established by previous models: if a feature $j$ has already been used by a prior model and was missing for a given observation $\boldsymbol{x}_i$, subsequent models can continue to rely on that feature without incurring additional penalty for $\boldsymbol{x}_i$. We control this by updating the weight $\sigma_{i,j}$ in (4) before fitting the next ensemble model,

$$\sigma'_{i,j} = \sigma_{i,j} \cdot (1 - \mathbb{1}[j \in \pi_{h_m}(\boldsymbol{x}_i)]\mathbb{1}[x_{i,j} = \mathbf{na}]) .$$

This ensures that subsequent models in the ensemble prefer to include features that have already been used by models from previous iterations, or features that incur low miss-

ingness reliance. Consequently, the ensemble can minimize prediction error while preventing potential excessive reliance on missing values from independently fit trees.

## 5. Balancing Missingness Reliance and Predictive Performance

One motivation for MA learning is that many model classes exhibit large Rashomon sets (Semenova et al., 2022)—collections of near-optimal models with similar predictive performance but differing feature properties. By adding a missingness reliance penalty to the learning objective, we prioritize models within this set that rely less on missing values. Still, important questions remain: When is a favorable trade-off between accuracy and missingness reliance possible? Are there non-trivial problems where both can be achieved without sacrificing the other? Are there problems where the MA learning objective is unsuitable? Next, we give a class of data generating processes (DGPs) where there exists a model $h$ that achieves both optimal predictive accuracy *and* zero missingness reliance, $\rho(h) = 0$. After that, we study cases where the MA approach may fail to improve over standard solutions for missing values.

### 5.1. Can We Achieve Both Zero Missingness Reliance and Minimal Prediction Error?

As an example where the heading's question holds, consider a system in which a variable is *never* missing under certain conditions but may be missing otherwise.

**Example 1.** Patients listed with a general healthcare provider undergo annual check-ups to assess their overall health. Certain patient data, such as age $(x_1)$, are always collected, while test results may be missing due to clinical recommendations and practitioner decisions. For example, cognitive tests are consistently administered to individuals over 65 years of age $(x_1 > 65)$, ensuring cognitive test results $(x_2)$ are available for all patients in this group. Additionally, patients with a positive test result undergo an MRI scan, which records hippocampal volume $(x_3)$ as above or below average. MRI scans may also be ordered for other reasons, such as spine problems or cartilage abnormalities.

We refer to such patterns in the data collection process as *observed deterministic data collection rules* (ODDC rules).

**Definition 1.** Suppose a distribution $p(X, M)$ over features $X \in \mathbb{R}^d$ and a missingness mask $M \in \{0, 1\}^d$ are given. Let $j$ be a variable, $T \subseteq [d] \setminus \{j\}$ a subset of other variables, and $A \subseteq \mathcal{X}_T$ be a subset of their domain $\mathcal{X}_T$. An ODDC rule $R = (T, A, j)$ is an implication in $p$ that whenever $X_T$ are observed and take values in $A$, variable $j$ is observed,

$$R : M_T = \mathbf{0}, X_T \in A \Longrightarrow_p M_j = 0 , \qquad (7)$$

or, equivalently, $p(M_j = 0 \mid M_T = \mathbf{0}, X_T \in A) = 1$. [1]

When fitting a prediction model to data, the learned structure can exploit ODDC rules to limit missingness reliance.

**Example 1 (Continued).** Suppose we want to fit a decision tree to the healthcare data to predict whether a patient suffers from cognitive impairment. Due to the ODDC structure, we can construct an accurate tree with zero missingness reliance by first splitting on the patient's age ($x_1$), followed by cognitive test results ($x_2$) for patients older than 65, as illustrated in Figure 1. For patients with $x_1 > 65$ and $x_2 = 1$, the MRI scan outcome—the most predictive variable—is always available and can be used for prediction.

We say that a node $u$ in a tree $h$ satisfies a set of ODDC rules $\mathcal{R} = \{R_1, ..., R_K\}$ if the feature $x_{j_u}$ used by node $u$ is implied to be observed by $\mathcal{R}$. Specifically, the truth values of the splits of $u$'s ancestors, $\mathbf{an}(u)$, must activate (imply the antecedent of) at least one rule $R \in \mathcal{R}$ that implies $M_{j_u} = 0$ according to (7). MA trees prefer node splits that satisfy ODDC rules by design: as $\alpha \to \infty$, only splits on variables that have no missingness for observations at the current node will be considered. A full tree $h$ satisfies a set of ODDC rules $\mathcal{R}$ if all of its internal nodes obey $\mathcal{R}$. Such trees have no reliance on missing values.

**Proposition 1.** *Let a set of ODDC rules $\mathcal{R}$ hold under a distribution $p(X, M)$. Any tree $h$ that satisfies $\mathcal{R}$ will have missingness reliance $\rho(h) = 0$ under $p$.*

Finally, we say that an outcome $Y$ satisfies a set of ODDC rules $\mathcal{R}$ with respect to an input $X$ and the missingness mask $M$ if $p(Y \mid X)$ is fully determined by variables implied to be observed by $\mathcal{R}$. We can conclude the following result.

**Corollary 1.** *Consider a distribution $p(X, M, Y)$. Suppose the outcome $Y$ satisfies a set of ODDC rules $\mathcal{R}$ in $p$. Then, there exists a prediction model $h$ with minimal risk and zero missingness reliance, $\rho(h) = 0$, on $p$.*

Proposition 1 and Corollary 1 are proven in Appendix A.

Corollary 1 follows by definition: if the outcome $Y$ can be described by input features known to be observed together, a model that fits $p(Y \mid X, M)$ perfectly can be optimally predictive and have zero reliance $\rho$. Such outcomes are more common in practice than we may first expect, at least approximately. For example, when predicting a medical diagnosis or mimicking a treatment policy, the ground-truth labels are given by decisions made by doctors based on observations they have made in sequence. Consequently, a tree-based model that matches test ordering patterns will use only information likely to be measured for splitting.

---

## 5.2. When Is Low Missingness Reliance Incompatible With Good Predictions?

The MA framework is designed to exploit structure in the missingness patterns induced by the data-generating process, as exemplified in the previous section. When there is no structure in $M$ predictable from observed variables $X$, the use of any feature $X_j$ will incur missingness reliance.

**Proposition 2.** *Suppose that each feature $j$ is independently missing completely at random with probability $p_j \in [0, 1]$,*

$$p(M_j \mid X^c, Y, M_{\neg j}) = p_j .$$

*Then, for any model $h$, with $a_h(\boldsymbol{x}, j) \in \{0, 1\}$ indicating whether the feature $x_j$ is used in the prediction $h(\boldsymbol{x})$,*

$$\rho(h) \geq \max_j \mathbb{E}[a_h(X, j)] p_j .$$

A proof is given in Appendix A. In words, the missingness reliance is lower-bounded by the frequency by which a feature is used by the model, multiplied by the missingness frequency, no matter the structure of the model. We give two more challenging cases for MA learning below.

**Distribution Shift.** If the test-time missingness distribution differs from the distribution of training data, deployment performance may degrade also for MA models, like prediction with missingness indicators or impute-then-regress strategies. For example, MA trees optimize decision paths based on observed missingness patterns, but if these change, a data-collection rule may no longer hold, and variables that were previously guaranteed to be observed may be missing test time. Both features frequently missing during training but available at test time and vice versa can lead to miscalibrated reliance and a suboptimal tradeoff with accuracy.

**MNAR and Informative Missingness.** In MNAR (missing not at random) settings, where the missingness mask depends on unobserved features, MA trees cannot strategically split to avoid missing values as there may be no discernible pattern in the missingness to influence the second term in (2). We study this effect empirically in Section 6.3. Moreover, when missingness is informative, i.e., when $P(M \mid Y) \neq P(M)$, MA learning may underperform compared to approaches that include missingness indicators, as these can become predictive in such cases. In principle, missingness indicators could be used with the MA penalty, but we refrain from exploring this here.

## 6. Experiments

We demonstrate the MA framework in a suite of experiments, aiming to show that MA models reduce reliance on

missing values while preserving predictive performance. We compare MA-DT, MA-LASSO, MA-GBT, and MA-RF to standard models as well as to models designed to handle missing values, using different imputation methods.

**Datasets.** We study six datasets with varying degrees of missingness. We sample 10,000 samples from the National Health and Nutrition Examination Survey (NHANES) for hypertension prediction with 42 features (Johnson et al., 2013). LIFE (2,864 samples) predicts whether life expectancy is above or below the median using 18 features (World Health Organization (WHO), 2021). ADNI (1,337 samples) predicts diagnosis changes in patients with suspected dementia using 39 features (Weiner et al., 2010). Breast Cancer (1,756 samples) includes 16 features for cancer prediction (Razavi et al., 2018). Pharyngitis (676 samples) is used for pharyngitis prediction using 18 features (Miyagi, 2023). Explainable Machine Learning Challenge dataset, referred to as FICO, consists of 10,549 samples and is used for credit risk classification with 23 features (FICO et al., 2018). Details on dataset preprocessing are in Appendix C.

**Baselines.** We compare MA models with baselines representing a) impute-then-regress learners, b) models using missingness indicators, and c) models that handle missing values natively. For a), we use $L^1$-regularized logistic regression (LR), a decision tree (DT), and a random forest (RF). For b), we consider M-GAM (McTavish et al., 2024), a sparse additive modeling approach that incorporates missingness indicators (ind) and their interactions with observed data (int). For c), we include XGBoost (Chen & Guestrin, 2016), NeuMiss (Le Morvan et al., 2020)—a neural network architecture with a nonlinearity based on missingness patterns in the input data—and MINTY (Stempfle & Johansson, 2024), a generalized linear rule model fit to minimize reliance on missing values. For models relying on imputation, we impute numerical features with zero imputation ($I_0$) and MICE (Van Buuren, 1999) ($I_M$). Categorical features are imputed with the mode.

**Experimental Setup.** We divide each dataset into training and testing subsets using an 80/20 split. For hyperparameter selection, including $\alpha$, we perform 3-fold cross-validation, evaluating candidate models based on a combination of the area under the receiver operating characteristic curve (AUROC) and empirical missingness reliance ($\hat{\rho}$). Specifically, we select the candidate model with the lowest $\hat{\rho}$ among those achieving at least $95$ percent of the maximum AUROC. This procedure is repeated for 5 different splits of the dataset, and we report $95\%$ confidence intervals based on the bootstrap distribution of AUROC and $\hat{\rho}$. Numerical features are standardized to have zero mean

and unit variance for all models except MINTY and M-GAM for which we apply discretization. Categorical features are encoded using one-hot encoding. All models are evaluated on the held-out test set with respect to AUROC and $\hat{\rho}$. Appendix C.3 provides details on the experimental setup, including hyperparameters. The code is available at https://github.com/Healthy-AI/malearn.

### 6.1. MA Models Match Baselines With Minimal Reliance on Missing Values

In Table 1, we report AUROC and missingness reliance ($\hat{\rho}$) for each model in ADNI, FICO, LIFE, and NHANES using zero imputation. Results using MICE imputation and for the other datasets, Pharyngitis and Breast Cancer, are included in Appendix D. Comparing MA-LASSO, MA-DT, and MA-RF to their unregularized counterparts, we find that confidence intervals for AUROC overlap in all cases except for the linear models on FICO. MA models demonstrate superior performance in avoiding missingness reliance, with an average decrease of 66.0 (MA-LASSO vs. LR), 4.4 (MA-DT vs. DT), and 70.2 (MA-RF vs. RF) across datasets in Table 1. MA-DT stands out by consistently having the lowest missingness reliance while maintaining competitive AUROC scores compared to DT. MA-GBT performs slightly worse compared to XGBoost in terms of AUROC but reduces missingness reliance by 64.4 on average.

MINTY is a strong baseline, with AUROC and missingness reliance comparable to MA-LASSO in most cases. However, its training time ranges from 18 to 292 seconds, whereas MA-LASSO trains in under a second. The M-GAM models exhibit moderate $\hat{\rho}$, suggesting that their use of the missingness structure leads to increased missingness reliance compared to MA models. NeuMiss achieves high AUROC but also exhibits high missingness reliance due to dense multiplications of input features.

The extreme differences observed in NHANES likely stem from its structured missingness patterns. Specifically, the missingness mask depends on the unobserved variable year, with different features and feature versions measured in different years. However, some measurements can act as substitutes for others. For example, there are several different features representing blood pressure measured using different equipment. These patterns cause many standard models to over-rely on missing features, whereas MA models effectively mitigate this issue. Interestingly, DT tends to avoid redundancy by selecting only the most informative features, whereas LR uses all available features, resulting in a very high missingness reliance.

### 6.2. Performance of MA Models in the Limits of $\alpha$

In addition to selecting $\alpha = \alpha^*$ based on the AUROC-missingness reliance trade-off, we fit models with $\alpha = 0$

*Table 1.* Test-set AUROC and missingness reliance ($\hat{\rho}$) across four datasets, using zero imputation. The 95 % confidence intervals are based on the bootstrap distribution of the respective metric. For each dataset and MA estimator, $\alpha$ is selected as part of the overall model selection strategy, with the model minimizing $\hat{\rho}$ chosen among those achieving at least 95 % of the maximum AUROC.

| | ADNI | | FICO | | LIFE | | NHANES | |
|---|---|---|---|---|---|---|---|---|
| ESTIMATOR | AUROC (↑) | $\hat{\rho}$ (↓) | AUROC (↑) | $\hat{\rho}$ (↓) | AUROC (↑) | $\hat{\rho}$ (↓) | AUROC (↑) | $\hat{\rho}$ (↓) |
| LR | 72.0 (69.7, 74.3) | 64.1 (62.2, 66.6) | 79.1 (78.3, 79.7) | 75.4 (74.9, 75.9) | 98.7 (97.7, 99.3) | 63.4 (60.6, 66.8) | 85.1 (84.6, 85.7) | 100.0 (100.0, 100.0) |
| DT | 72.7 (71.4, 74.2) | 11.7 (6.3, 18.8) | 73.8 (73.0, 74.5) | 5.7 (5.4, 6.0) | 92.2 (90.1, 93.6) | 18.3 (15.8, 19.9) | 82.3 (81.5, 83.0) | 0.1 (0.0, 0.2) |
| RF | 75.0 (72.9, 77.0) | 66.1 (64.3, 68.3) | 77.3 (76.4, 78.1) | 65.4 (64.6, 66.6) | 97.8 (95.8, 99.0) | 82.8 (81.8, 83.9) | 81.6 (81.4, 81.9) | 100.0 (100.0, 100.0) |
| M-GAM (IND) | 69.7 (66.9, 72.1) | 14.7 (0.6, 32.5) | 77.9 (77.6, 78.3) | 42.4 (28.6, 56.2) | 98.2 (96.4, 99.2) | 49.4 (42.8, 56.1) | 84.0 (83.4, 84.6) | 28.2 (9.5, 50.0) |
| M-GAM (INT) | 69.5 (67.0, 72.0) | 46.3 (22.5, 61.2) | 77.6 (77.3, 77.9) | 29.1 (28.1, 30.2) | 98.1 (96.3, 99.3) | 51.7 (45.3, 60.2) | 84.1 (83.6, 84.5) | 26.4 (14.8, 48.3) |
| XGBOOST | 79.0 (75.7, 82.4) | 60.9 (57.6, 63.4) | 78.4 (77.9, 79.0) | 69.3 (58.9, 74.9) | 98.2 (96.1, 99.4) | 56.1 (52.3, 61.1) | 84.8 (84.6, 85.2) | 95.7 (87.3, 100.0) |
| NEUMISS | 76.4 (75.0, 78.2) | 67.0 (64.9, 70.1) | 79.0 (78.4, 79.5) | 75.4 (74.9, 75.9) | 98.6 (97.3, 99.4) | 84.1 (83.0, 85.2) | 85.5 (85.2, 85.7) | 100.0 (100.0, 100.0) |
| MINTY | 70.6 (67.4, 73.8) | 1.3 (0.3, 2.5) | 76.6 (75.2, 77.9) | 22.8 (8.0, 43.6) | 96.2 (94.1, 98.3) | 21.9 (16.8, 27.2) | 81.9 (80.1, 83.8) | 0.3 (0.0, 0.8) |
| MA-LASSO | 68.4 (66.1, 70.6) | 11.8 (0.1, 34.7) | 75.8 (74.9, 76.6) | 5.7 (5.4, 6.0) | 97.7 (96.7, 98.6) | 21.0 (19.1, 24.1) | 84.5 (84.0, 85.0) | 0.4 (0.0, 0.9) |
| MA-DT | 74.1 (71.7, 76.4) | 0.1 (0.0, 0.3) | 73.8 (73.0, 74.5) | 5.7 (5.4, 6.0) | 89.7 (87.5, 91.8) | 12.3 (10.4, 15.3) | 82.4 (81.6, 83.2) | 0.0 (0.0, 0.0) |
| MA-RF | 78.0 (76.2, 80.4) | 3.2 (0.7, 7.6) | 76.3 (75.6, 76.8) | 5.8 (5.5, 6.1) | 97.2 (95.7, 98.5) | 24.2 (22.0, 26.5) | 84.0 (83.3, 84.6) | 0.2 (0.0, 0.5) |
| MA-GBT | 78.5 (76.4, 80.5) | 1.6 (0.7, 2.7) | 75.6 (74.5, 77.1) | 5.9 (5.5, 6.2) | 95.9 (92.1, 98.9) | 16.5 (11.9, 21.2) | 83.2 (82.1, 84.4) | 0.3 (0.0, 1.0) |

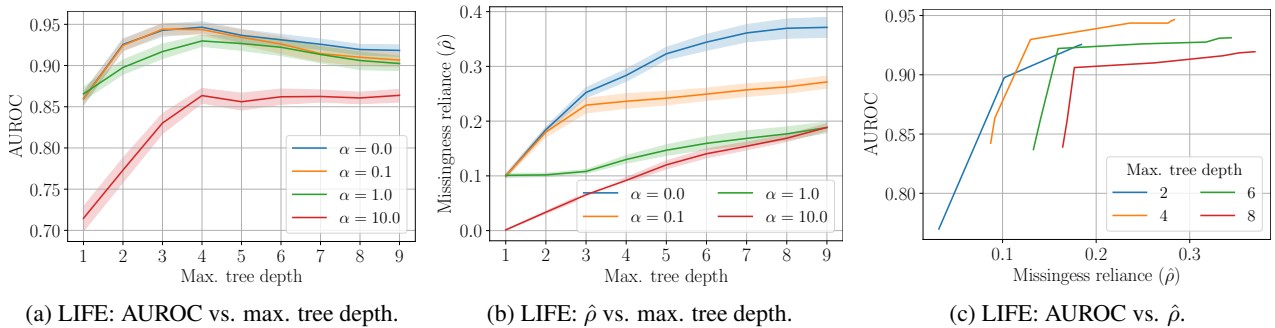

(a) LIFE: AUROC vs. max. tree depth.  (b) LIFE: $\hat{\rho}$ vs. max. tree depth.  (c) LIFE: AUROC vs. $\hat{\rho}$.

*Figure 3.* Average test performance across cross-validation folds for MA decision trees fit to LIFE using varying values of $\alpha$ and `max_depth`. A shallow tree of depth 3–4 captures the complexity of the data effectively. Missingness regularization with $\alpha \leq 1.0$ does not significantly reduce predictive performance but helps decrease missingness reliance from approximately 30 % to approximately 10 %.

and $\alpha$ set high enough to ensure $\hat{\rho} \approx 0$ while maximizing AUROC (denoted as $\alpha = \infty$). Following scikit-learn's convention, a tree with `max_depth = 1` is allowed one split if a meaningful feature is available. Consequently, achieving $\hat{\rho} \approx 0$ may be impossible when all features contain missing values, preventing full elimination of missingness reliance in tree-based models. We report AUROC and $\hat{\rho}$ for each setting and dataset in Table 6 and 7 in Appendix D. Figure 4a and 4b show how AUROC and $\hat{\rho}$ change when transitioning from $\alpha = \infty$ to $\alpha = 0$ for ADNI. Here, all MA models achieve $\hat{\rho} \approx 0$ with a large $\alpha$. While setting $\alpha = 0$ can significantly boost AUROC, it drastically increases missingness reliance, especially for `MA-LASSO` and `MA-RF`.

We further examine the impact of increasing missingness regularization when fitting `MA-DT` with varying values for the `max_depth` parameter. Figure 3 and Figure 5 in Appendix D illustrate the trends in AUROC and missingness reliance for LIFE. When $\alpha = 0$, `MA-DT` behaves as a standard decision tree without missingness reliance regularization. For $\alpha \in \{0, 0.1, 1\}$, a depth of 3–4 captures the data's complexity without overfitting. On the other hand, $\hat{\rho}$ varies

from 0.1 ($\alpha = 1$) to 0.3 ($\alpha = 0$), demonstrating a clear case where missingness regularization can be applied without sacrificing performance. Notably, reducing the tree's maximum depth is a less effective way to control missingness reliance, as it negatively impacts AUROC.

In Figure 6 in Appendix D, we show examples of MA trees with $\alpha \in \{0, \alpha^*, \infty\}$ at their respective optimal depths for LIFE. When $\alpha = \infty$, the learning algorithm is forced to split on the always-observed feature `Region` instead of `Adult_mortality`, which has around 10 % missingness and is used when $\alpha \in \{0, \alpha^*\}$. With $\alpha = \alpha^*$, `Region` appears in the first split in the right branch of the tree, reducing missingness reliance without sacrificing AUROC compared to the unregularized tree, which uses the incomplete feature `Under_five_deaths`.

### 6.3. Performance of MA Models Across Missingness Mechanisms

To assess MA models under different missingness mechanisms, we introduce 50 % synthetic missingness to randomly selected features in the Breast Cancer dataset, the

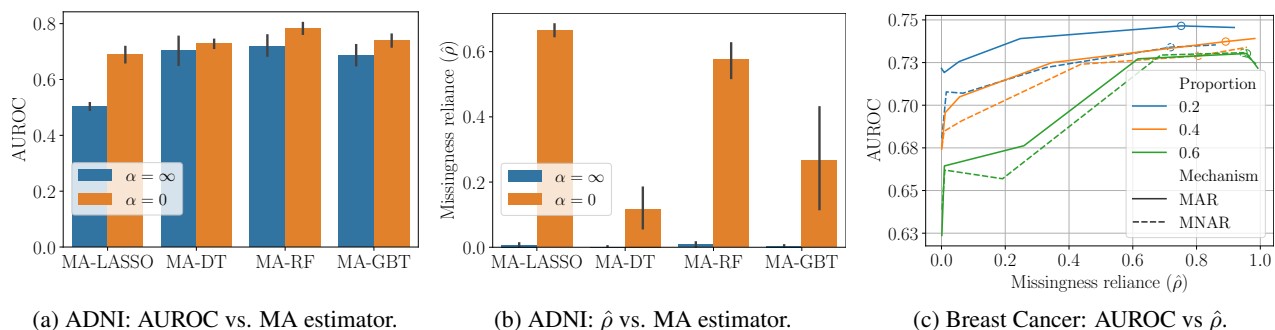

(a) ADNI: AUROC vs. MA estimator.  (b) ADNI: $\hat{\rho}$ vs. MA estimator.  (c) Breast Cancer: AUROC vs $\hat{\rho}$.

*Figure 4.* (a) and (b): Test-set AUROC and missingness reliance ($\hat{\rho}$) for MA estimators in ADNI when transitioning from $\alpha = \infty$ to $\alpha = 0$. Removing missingness regularization improves predictive performance, but the increase in missingness reliance is much more pronounced, especially for `MA-LASSSO` and `MA-RF`. (c): Test-set AUROC versus $\hat{\rho}$ for `MA-LASSSO` in Breast Cancer, where $50\,\%$ synthetic missingness is added to an increasing proportion of input features. Missingness not at random (MNAR) is more challenging than missingnes at random (MAR), but `MA-LASSSO` demonstrates robust performance for large fractions of missingness.

most complete in our study. We apply MAR using a logistic masking model and MNAR for values below the 0.25 and above the 0.75 quantiles. Figure 4c shows how AUROC varies with missingness reliance for different missingness proportions and mechanisms using `MA-LASSO`. The performance of $L^1$-regularized logistic regression models is included with circles for reference. When MAR missingness is introduced in up to $40\,\%$ of the features, it is possible to fit `MA-LASSO` with $25\,\%$ missingness reliance without sacrificing much predictive performance. Notably, the trends are similar for the more challenging MNAR case, although the curves are slightly shifted toward lower AUROCs in all cases.

# 7. Discussion

We have proposed a missingness-avoiding (MA) machine learning, a framework for reducing the reliance of models on missing values in prediction, applied to sparse linear models (`MA-LASSO`), decision trees (`MA-DT`), random forests (`MA-RF`), and gradient boosting (`MA-GBT`). Our experiments show that MA models effectively learn to generate accurate predictions while minimizing reliance on missing values at test time by applying classifier-specific regularization to avoid using unobserved feature values.

To apply the MA framework, begin by selecting a model class that suits the interpretability needs of the application. Linear models and decision trees offer more transparency, while ensembles often provide better performance with less interpretability. After choosing a model type, adjust the regularization parameter $\alpha$ to balance predictive performance, and reliance on missing features. A higher $\alpha$ discourages reliance on frequently missing features, while a lower $\alpha$ favors performance. The right value depends on the acceptable level of missingness reliance. Practitioners can evaluate their strategy using the missingness reliance

metric $\rho$, which measures the model's dependence on missing features.

As a future direction for tree-based MA models, an alternative to minimize reliance on missing values could be to enforce $\hat{\rho} = 0$ at test time by adopting a fallback strategy—halting the decision process and returning the label of the current node whenever a missing value is encountered. Additionally, pruning strategies and hyperparameter tuning beyond depth constraints could be explored to reduce missingness reliance in these models. In our current implementation of `MA-DT`, we rely on greedy splitting algorithms, but in theory, global optimization approaches (Gurobi Optimization, LLC, 2024; ILOG, IBM, 2010) could also be regularized with missingness reliance, producing accurate decision trees with potentially improved interpretability (Rudin et al., 2022).

Moreover, further analysis could explore alternative strategies for selecting $\alpha^*$ beyond our current approach and investigate different definitions of $\rho$, as discussed in Section 3. Alternative strategies for selecting $\alpha^*$ should consider the specific application needs. For instance, rather than focusing solely on the acceptable trade-off in predictive performance for achieving the lowest possible $\alpha$, we might prioritize limiting the number of missing features per individual. Relying on an average measure across the dataset may not provide sufficient insight. Additionally, domain knowledge can help define an acceptable level of missingness, but quantifying this threshold is often challenging and subjective. Our focus has been on the tradeoff between predictive accuracy and reliance on missing values in prediction tasks that match the training data in distribution. Future work should explore other effects of MA learning, such as robustness to distributional shifts. Lastly, an open question remains: can our method be extended to other model classes?

## Acknowledgements

This work was partially supported by the Wallenberg AI, Autonomous Systems and Software Program (WASP) funded by the Knut and Alice Wallenberg Foundation.

The computations and data handling were enabled by resources provided by the National Academic Infrastructure for Supercomputing in Sweden (NAISS), partially funded by the Swedish Research Council through grant agreement no. 2022-06725.

Data used in the preparation of this article were obtained from the Alzheimer's Disease Neuroimaging Initiative (ADNI) database (https://adni.loni.usc.edu/). The ADNI was launched in 2003 as a public-private partnership, led by Principal Investigator Michael W. Weiner, MD. The primary goal of ADNI has been to test whether serial magnetic resonance imaging (MRI), positron emission tomography (PET), other biological markers, and clinical and neuropsychological assessment can be combined to measure the progression of mild cognitive impairment (MCI) and early Alzheimer's disease (AD).

## Impact Statement

Two of our proposed model classes, `MA-DT` and `MA-LASSO`, prioritize interpretability to ensure robust and practitioner-friendly models. This focus on interpretability is crucial for applying machine learning in high-stakes domains like healthcare, where handling missing data poses risks of introducing or reinforcing unfairness (Jeanselme et al., 2022). Through increased transparency, these models aim to support ethical decision-making in complex environments.

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

# A. Proof of Propositions 1, 2 and Corollary 1

**Proposition 1 (Restated).** Let a set of ODDC rules $\mathcal{R}$ hold under a distribution $p(X, M)$. Any tree $h$ that satisfies $\mathcal{R}$ will have missingness reliance $\rho(h) = 0$ under $p$.

*Proof.* We will prove the proposition by induction on the size (number of nodes) $n$ of the decision tree $h$.

BASE CASE: TREE SIZE $n = 0$ (TREE WITH ONLY A ROOT NODE)

Let $h$ be a decision tree with size $n = 0$ that obeys the ODDC rules $\mathcal{R}$. The decision tree $h$ consists solely of a root node and it is also a leaf node. Since there are no splits, no features are used in computing $h(\boldsymbol{x})$. Therefore, for any observation $\boldsymbol{x} \in \mathcal{X}$, $\rho(h, \boldsymbol{x}) = 0$. Consequently, the expected missingness reliance is:

$$\rho(h) = \mathbb{E}_p\left[\rho(h, X)\right] = \mathbb{E}_p[0] = 0.$$

Thus, the base case holds.

INDUCTION STEP: SIZE $n = d + 1$

Assume that for any decision tree $h$ of size $d$ that satisfies the ODDC rules in $\mathcal{R}$, the expected missingness reliance satisfies:

$$\rho(h) = 0.$$

Now, consider a decision tree $h'$ of size $d + 1$, built from a tree $h$ of size $d$ by adding a single split $(j, \tau)$ to a leaf node $\ell \in \mathcal{L}(h)$. Assume that $h'$ also satisfies $\mathcal{R}$. Since $h$ satisfies the ODDC rules, $\rho(h) = 0$ by the induction assumption.

By Equation (3), splitting a leaf $\ell \in \mathcal{L}(h)$ using feature and threshold $(j, \tau)$ can only introduce missingness reliance $\rho(h') > 0$ if the feature $x_j$ is missing for some input $\boldsymbol{x}$ such that $\ell \in \pi_h(\boldsymbol{x})$ ($\ell$ is on the decision path for $\boldsymbol{x}$ in $h$). However, if splitting on feature $j$ could lead to $x_j = \mathbf{na}$ for any such $\boldsymbol{x}$, the resulting tree $h'$ would not satisfy $\mathcal{R}$, and we would have a contradiction. Given that $h'$ satisfies $\mathcal{R}$, it must be that

$$\forall \boldsymbol{x} : \ell \in \pi_h(\boldsymbol{x}) \; ; \; p(M_j = 1 \mid X = \boldsymbol{x}) = 0 \,.$$

Thus, the new tree $h'$ also has zero missingness reliance, $\rho(h') = 0$. Since the base case holds and the inductive step has been proven, by mathematical induction, by the inductive hypothesis, for trees $h$ of any size $n$ that satisfy the ODDC rules $\mathcal{R}$, the missingness reliance is $\rho(h) = 0$. $\qquad\square$

**Corollary 1 (Restated).** Consider a distribution $p(X, M, Y)$. Suppose the outcome $Y$ satisfies a set of ODDC rules $\mathcal{R}$ in $p$. Then, there exists a prediction model $h$ with minimal risk and zero missingness reliance, $\rho(h) = 0$, on $p$.

*Proof.* We can prove the corollary constructively by defining a prediction model with minimal risk on $p$. For any loss function $L$, there is a (Bayes optimal) classifier $h^*$ which minimizes the expected risk over $p$ by making point-wise optimal predictions,

$$\forall \boldsymbol{x} : h^*(\boldsymbol{x}) := \underset{y : p(y|x) > 0}{\arg\min} \; \mathbb{E}[L(y, Y) \mid X = x], \tag{8}$$

and therefore,

$$\underset{h : \mathcal{X} \to \mathcal{Y}}{\arg\min} \; \mathbb{E}_X\left[\mathbb{E}_{Y|X}[L(h(X), Y) \mid X]\right] = h^* \,.$$

For the squared loss function $L(y, y') = (y - y')^2$, $h^*$ is the conditional expectation, $h^*(\boldsymbol{x}) = \mathbb{E}[Y \mid X = \boldsymbol{x}]$.

Since $Y$ satisfies $\mathcal{R}$ on $p$ by assumption, the function $f(x) = p(Y = y \mid \boldsymbol{x})$ itself has missingness reliance $\rho(f) = 0$ on $p$. Since $h^*$ can be computed deterministically from $p(Y = y \mid X = \boldsymbol{x})$, $\rho(h^*) = 0$. Therefore, $h^*$ has both minimal risk and zero missingness reliance on $p$. $\qquad\square$

**A Note on the Hypothesis Class.** In Corollary 1, the optimal model $h^* = \mathbb{E}[Y \mid X]$ is implicitly assumed to belong to the hypothesis class $\mathcal{H}$. While this assumption is benign for universal function approximators like decision trees, it becomes more consequential for more restrictive model classes, such as linear models, where $h^*$ may lie outside $\mathcal{H}$.

**A Note on Convergence.** If an optimal model $h^*$ exists, can our proposed algorithms recover it given enough data? The answer depends on the model class, the optimization procedure, and the data distribution. For instance, if $h^*$ is a sparse logistic model, then solving the optimization problem (5) using the $L^1$-based regularization is likely to recover it—similar to how standard Lasso behaves.

**Proposition 2 (Restated).** Assume that each feature $j$ is independently missing completely at random with probability $p_j$,

$$p(M_j \mid X, Y, M_{\neg j}) = p_j .$$

Then, for any model $h$, with $a_h(\boldsymbol{x}, j) \in \{0, 1\}$ indicating whether feature $x_j$ is used in the prediction $h(\boldsymbol{x})$,

$$\rho(h) \geq \max_j \mathbb{E}[a_h(X, j)] p_j .$$

*Proof.* By definition, we have that

$$\rho(h) = \mathbb{E}[\max_j a_h(X, j) \mathbb{1}[X_j = \mathbf{na}]] .$$

It follows that

$$
\begin{aligned}
\rho(h) &\geq \max_j \mathbb{E}[a_h(X, j) \mathbb{1}[M_j = 1]] \\
&= \max_j \mathbb{E}[a_h(X, j)] \mathbb{E}[M_j = 1] \\
&= \max_j \mathbb{E}[a_h(X, j)] p_j ,
\end{aligned}
$$

where the second row follows due to independence between $M_j$ and $X$. $\qquad\square$

## B. Missingness-Avoiding Gradient Boosting

In Algorithm 1, we present the algorithm used to fit `MA-GBT`.

---

**Algorithm 1** Missingness-Avoiding Gradient Boosted Trees (MA-GBT)

---

1: **inputs** Data set $\mathcal{D} = \{(\boldsymbol{x}_i, y_i)\}_{i=1}^n$, learning rate $\gamma \geq 0$, regularization parameter $\alpha$, loss function $L$
2: **inputs** Ensemble size $M$
3: $e_0(\boldsymbol{x}) = \arg\min_c \sum_{i=1}^n L(y_i, c)$
4: $\forall i \in [n], j \in [d] : \sigma_{i,j} = 1$
5: **for** $m = 1, \ldots, M$ **do**
6:     Compute pseudo-residuals:
        $\forall i \in [n] : r_{im} = -\left[\frac{\partial L(y_i, e(\boldsymbol{x}_i))}{\partial e(\boldsymbol{x}_i)}\right]_{e(\boldsymbol{x}) = e_{m-1}(\boldsymbol{x})}$
7:     Fit MA regression tree $h_m$ to $\mathcal{D}_r = \{(\boldsymbol{x}_i, r_{im})\}_{i=1}^n$ using the splitting objective in (4) with regularization $\alpha$,
8:         sample multipliers $\{\{\sigma_{i,j}\}_{j=1}^d\}_{i=1}^n$ and mean squared error as the splitting criterion $C$
9:     Update ensemble:
10:         $e_m(\boldsymbol{x}) = e_{m-1}(\boldsymbol{x}) + \gamma h_m(\boldsymbol{x})$
11:     Update reliance weights:
12:         $\forall i \in [n], j \in [d] : \sigma_{i,j} = \sigma_{i,j} \cdot (1 - \mathbb{1}[j \in \pi_{h_m}(\boldsymbol{x}_i)] \mathbb{1}[x_{i,j} = \mathbf{na}])$
13: **end for**

---

## C. Experimental Details

In this section, we provide additional details about our experimental setup.

### C.1. Datasets

In Table 2, we provide an overview of the datasets used in our experiments. Each dataset is described in more detail below. The Life, Breast Cancer, and Pharyngitis datasets are available under a Creative Commons license. The FICO dataset is

subject to its own licensing terms, which can be accessed at: https://community.fico.com/s/explainable-machine-learning-challenge. The ADNI dataset requires agreement to the data use policy available at: ADNI Data Use Agreement. The NHANES dataset is governed by the data release policy accessible here: NHANES Data Release Policy.

For preprocessing the datasets, we followed the structure from Shadbahr et al. (2023). For the synthetic missingness experiment described in Section 6.3, we adopted the approach from Mayer et al. (2022). Additional preprocessing details can be found on our GitHub project page: https://github.com/Healthy-AI/malearn.

Table 2. Overview of the datasets used in our experiments. Each dataset corresponds to a binary classification task.

| Dataset | Samples | Features | Prediction Task |
|---|---|---|---|
| NHANES (Johnson et al., 2013) | 69,118 | 42 | Hypertension |
| LIFE (World Health Organization (WHO), 2021) | 2,864 | 18 | Life expectancy above/below dataset median |
| ADNI (Weiner et al., 2010) | 1,337 | 39 | Diagnosis change over two years |
| Breast Cancer (Razavi et al., 2018) | 1,756 | 16 | Hormone receptor status |
| Pharyngitis (Miyagi, 2023) | 676 | 18 | Outcome of rapid antigen detection test |
| FICO (FICO et al., 2018) | 10,549 | 23 | Credit repayment |

**NHANES.** The National Health and Nutrition Examination Survey (NHANES) (Johnson et al., 2013) is a cross-sectional study designed to assess the health and nutritional status of the U.S. population. It combines interviews, physical examinations, and laboratory tests to provide a comprehensive dataset for epidemiological research. While NHANES is not inherently structured for a specific machine learning task, it has been widely used in predictive modeling studies (Gardner et al., 2023). In this work, NHANES was utilized to develop models for predicting hypertension, where the outcome variable ($Y$) wa based on clinical blood pressure measurements. The predictor variables ($X$) included demographic information such as age, sex, and ethnicity, examination features like BMI and waist circumference, laboratory values including cholesterol levels and blood glucose, as well as questionnaire-based factors such as smoking status and physical activity. Missingness in NHANES arises due to various factors, including participant refusal, ineligibility, and incomplete examinations. Participants may choose not to answer survey questions or refuse to undergo certain medical tests. Some components are only administered to specific demographic groups, such as dietary recalls that apply only to adults, resulting in structurally missing values. Additionally, missing data may occur due to technical issues or participant dropout, making certain variables unavailable for some individuals. Notably, missing values due to ineligibility are deterministic, meaning they cannot be reliably imputed as they are structurally absent in specific subgroups. For our experiments, we sampled 10,000 observations with equal distribution of positive and negative cases.

**Life Expectancy (WHO).** The Life Expectancy dataset, sourced from the World Health Organization (WHO) (World Health Organization (WHO), 2021) and available at https://www.kaggle.com/datasets/lashagoch/life-expectancy-who-updated aggregates country-level health indicators spanning from 2000 to 2015. It includes a diverse range of demographic, economic, and health-related variables that are used to predict life expectancy through both regression and classification tasks. This dataset comprises 2864 records across 179 countries and contains indicators such as population growth, fertility rates, and median age as demographic variables, GDP per capita and government healthcare expenditure as economic indicators, and immunization coverage, prevalence of diseases, and mortality rates as health system data. In this work, we created a classification task to determine whether a country's life expectancy is above or below the dataset median. Starting from the complete dataset obtained via the linked Kaggle project, we followed the preprocessing steps described in (Stempfle et al., 2023), introducing $10\%$ MCAR missingness across features. Observations with missing country information were excluded, as the train-test split was performed with respect to this variable (which was not included in the feature set). Missing values in the Region variable were imputed based on the corresponding country. The final dataset is available in our GitHub repository: https://github.com/Healthy-AI/malearn.

**Alzheimer's Disease Neuroimaging Initiative (ADNI).** The Alzheimer's Disease Neuroimaging Initiative (ADNI) (Weiner et al., 2010) is a longitudinal study designed to collect and standardize clinical, neuroimaging, and genetic data to support the early diagnosis and monitoring of Alzheimer's disease progression. The dataset provides detailed patient assessments at multiple time points, making it a valuable resource for predictive modeling. In this work, baseline data from a sample of 1,337 patients was used to construct a classification task: predicting whether a patient's diagnosis will

change two years after the baseline assessment. The selected dataset includes clinical data (e.g., cognitive test scores and medical history), neuroimaging data from MRI and PET scans (measuring brain atrophy and amyloid-beta accumulation), and genetic information, including APOE genotype, which is strongly associated with Alzheimer's risk. Missingness in ADNI arises from inconsistent follow-up visits, participant withdrawal, and variability in the availability of imaging or genetic tests. Since not all participants complete every assessment at each time point, missing data can significantly impact longitudinal analyses.

**Breast Cancer.** The Breast Cancer dataset, introduced by (Razavi et al., 2018), focuses on predicting hormone receptor (HR) status and survival time using molecular and clinical features. In this work, we focused on a classification task aimed at distinguishing between HR-positive (HR+) and HR-negative (HR−) tumors, as HR status is a key biomarker guiding treatment decisions—particularly the use of endocrine therapies. This task leverages features such as estrogen receptor (ER) and progesterone receptor (PR) status, HER2 expression, and mutation burden. The dataset contains 1,711 samples and 18 features, with a class distribution of 1,415 HR-positive and 296 HR-negative cases. Three variables exhibit missing values: "ER PCT Primary" (185 missing entries), "Fraction Genome Altered" (22), and "Mutation Count" (75). We followed the preprocessing approach described in McTavish et al. (2024), including only a single row per individual patient.

**FICO.** The FICO Explainable Machine Learning Challenge dataset (FICO et al., 2018) contains anonymized credit data from Home Equity Line of Credit (HELOC) applications, with the primary objective of predicting whether an individual will repay a line of credit within two years. The dataset includes 10,459 applicants and 23 financial attributes derived from credit reports, such as credit utilization ratios, payment history, and debt-to-income ratios, making it a valuable dataset for credit risk assessment. A unique feature of this dataset is its encoding of missing values through three distinct codes: $-7$ denotes missing information of a specific type, $-8$ indicates the complete absence of usable information, and $-9$ signifies cases where a credit bureau report was either not retrieved or not found. These encodings reflect real-world financial uncertainty and require tailored imputation strategies. The presence of structured missing values poses an additional challenge in developing models that are both interpretable and robust in assessing credit risk. In our preprocessing of the data, we replaced all the different missingness codes with **na**.

**Pharyngitis.** The Pharyngitis dataset, introduced by Miyagi (2023), consists of 676 patients with 19 measured features and was used to predict the outcome of a rapid antigen detection test (RADT) in children with pharyngitis. Specifically, the outcome was the presence of group A streptococcus (GAS) on throat culture. The predictor variables include age, sudden onset of sore throat, maximum body temperature (as reported by the accompanying parent), throat pain, cough, rhinorrhea, conjunctivitis, headache, pharyngeal erythema, tonsillar swelling, tonsillar exudate, palatal petechiae, nausea and/or vomiting, abdominal pain, diarrhea, tender cervical lymph nodes, and the presence of a scarlatiniform rash. The number of missing values ranged from $0\%$ to around $6\%$ per predictor variable.

### C.2. Hyperparameters

The hyperparameters we considered are shown in Table 3. We performed an exhaustive search over the hyperparameters for all models except `MA-RF`, `MA-GBT`, MINTY, and XGboost, for which we randomly sampled 10 different sets of hyperparameters. Note, in our implementation of `MA-LASSO` $\lambda_j = \left( \frac{\sum_{i=1}^{n} m_{i,j} + \beta}{n} \right) \alpha$, where $\beta = [0.001, 0.01, 0.1, 1, 10, 100, 1000]$.

### C.3. Missingness Reliance Metric for Each Estimator

The missingness reliance metric quantifies a model's dependence on missing values during inference, providing an estimator-specific yet comparable measure. We define missingness reliance for all MA models in Section 4. For tree-based baseline models (DT, RF, and XGBoost), this reliance is defined in the same way as for the corresponding MA model. For linear baseline models, such as LR and M-GAM, reliance is determined by the use of missing features with non-zero coefficients, i.e., in the same way as for `MA-LASSO`. NeuMiss explicitly incorporates missingness through mask non-linearity and specialized blocks, with reliance assessed by the proportion of samples requiring such mechanisms. In MINTY, reliance is defined at the rule level, indicating whether a missing attribute influences a given decision.

*Table 3.* Hyperparameters and their respective search space for all models.

| Model | Hyperparameter | Search space |
|---|---|---|
| LR | regularization strength | $\{0.1, 0.5, 1.0, 2.0, 10.0\}$ |
| DT | max depth | $\{1, 2, 3, 4, 5, 6, 7, 8, 9\}$ |
| RF | max depth
min samples to split | $\{3, 4, 5, 6, 7, 8, 9\}$
$\{0.05, 0.10, 0.15, 0.20, 0.25\}$ |
| M-GAM | lambda 0 | $\{20, 10, 5, 2, 1, 0.5, 0.4, 0.2, 0.1, 0.05, 0.02, 0.01, 0.005\}$ |
| XGBoost | max depth
learning rate
number of estimators | $\{3, 4, 5, 6, 7, 8, 9\}$
$\{0.01, 0.1\}$
$\{100, 200, 300, 400, 500\}$ |
| NeuMiss | learning rate
batch size | $\{0.001, 0.01, 0.1\}$
$\{32, 64, 128\}$ |
| MINTY | optimizer
max rules
lambda 0
lambda 1
gamma | $\{\text{"beam"}\}$
$\{10, 15\}$
$\{1.0 \times 10^{-6}, 1.0 \times 10^{-5}, 0.001, 0.01, 0.1, 10\}$
$\{1.0 \times 10^{-6}, 1.0 \times 10^{-5}, 0.001, 0.01, 0.1, 10\}$
$\{0, 0.001, 0.01, 0.1, 10000\}$ |
| MA-LASSO | alpha
beta | $\{1, 10, 100, 1000, 10000\}$
$\{0.001, 0.01, 0.1, 1, 10, 100, 1000\}$ |
| MA-DT | max depth
alpha | $\{1, 2, 3, 4, 5, 6, 7, 8, 9\}$
$\{0.001, 0.01, 0.1, 1, 10\}$ |
| MA-RF | number of estimators
max depth
alpha | $\{50\}$
$\{1, 2, 3, 4, 5, 6, 7\}$
$\{0.001, 0.01, 0.1, 1, 10\}$ |
| MA-GBT | number of estimators
loss
learning rate
max depth
alpha | $\{10\}$
$\{\text{"log\_loss"}\}$
$\{0.01, 0.1\}$
$\{1, 2, 3, 4, 5, 6, 7\}$
$\{0.001, 0.01, 0.1, 1, 10\}$ |

## C.4. Implementations

For Neumiss we used PyTorch (Paszke et al., 2019) in combination with skorch (Tietz et al., 2017). Model parameters were optimized using the cross-entropy loss and the Adam optimizer with default parameters. Early stopping was applied during training if there was no improvement in performance for 10 consecutive epochs. The logistic regression model, the decision tree classifier and the random forest classifier were implemented using the scikit-learn library (Pedregosa et al., 2011). For XGBoost, we used the XGBoost Python Package (see `https://xgboost.readthedocs.io/en/stable/python/index.html`). For MINTY and M-GAM, we adapted code provided by Stempfle & Johansson (2024) and McTavish et al. (2024). Our code is available at `https://github.com/Healthy-AI/malearn`.

NODE SPLITTING CRITERION FOR TREE-BASED MODELS

In our tree-bases models, we performed node splitting based on the Gini criterion, defined as:

$$C(\ell, \mathcal{D}; j, \tau) = \sum_c p_c^\ell (1 - p_c^\ell), \;\; p_c^\ell = \sum_{i \in \mathcal{S}_\ell} \frac{\mathbb{1}[y_i = c]}{|\mathcal{S}_\ell|},$$

for the general case with $c$ classes. In our case, $c = 2$. For definitions of additional notations, please refer to Section 4.

## D. Additional Results

In this section, we provide additional results from our experiments. Table 4 presents the same results as Table 1, but using MICE imputation instead of zero imputation. Table 5 shows results using both imputation techniques on two additional datasets: Breast Cancer and Pharyngitis. Table 6 and Table 7 report the performance of MA models under different settings

*Table 4.* Test-set AUROC and missingness reliance ($\hat{\rho}$) across four datasets using MICE imputation. The 95 % confidence intervals are based on the bootstrap distribution of the respective metric. For each dataset and MA estimator, $\alpha$ is selected as part of the overall model selection strategy, with the model minimizing $\hat{\rho}$ chosen among those achieving at least 95 % of the maximum AUROC.

| | ADNI | | FICO | | LIFE | | NHANES | |
|---|---|---|---|---|---|---|---|---|
| ESTIMATOR | AUROC (↑) | $\hat{\rho}$ (↓) | AUROC (↑) | $\hat{\rho}$ (↓) | AUROC (↑) | $\hat{\rho}$ (↓) | AUROC (↑) | $\hat{\rho}$ (↓) |
| LR | 72.9 (70.6, 75.1) | 62.9 (60.5, 64.7) | 79.1 (78.3, 79.6) | 75.4 (74.9, 75.9) | 99.0 (98.2, 99.6) | 54.9 (52.5, 57.1) | 85.1 (84.6, 85.7) | 100.0 (100.0, 100.0) |
| DT | 69.8 (67.6, 72.1) | 34.6 (25.0, 44.6) | 73.7 (73.0, 74.5) | 5.7 (5.4, 6.0) | 91.0 (87.9, 94.4) | 15.8 (12.0, 19.4) | 82.4 (81.7, 83.1) | 0.1 (0.0, 0.2) |
| RF | 76.6 (73.9, 79.8) | 65.9 (64.0, 68.1) | 77.5 (76.9, 78.1) | 71.0 (67.3, 74.3) | 97.9 (96.4, 98.7) | 83.6 (82.6, 84.7) | 82.3 (82.0, 82.5) | 100.0 (100.0, 100.0) |
| M-GAM (IND) | 69.7 (66.9, 72.1) | 14.7 (0.6, 32.5) | 77.9 (77.6, 78.3) | 42.4 (28.6, 56.2) | 98.2 (96.4, 99.2) | 49.4 (42.8, 56.1) | 84.0 (83.4, 84.6) | 28.2 (9.5, 50.0) |
| M-GAM (INT) | 69.5 (67.0, 72.0) | 46.3 (22.5, 61.2) | 77.6 (77.3, 77.9) | 29.1 (28.1, 30.2) | 98.1 (96.3, 99.3) | 51.7 (45.3, 60.2) | 84.1 (83.6, 84.5) | 26.4 (14.8, 48.3) |
| XGBOOST | 79.0 (75.7, 82.4) | 60.9 (57.6, 63.4) | 78.4 (77.9, 79.0) | 69.3 (58.9, 74.9) | 98.2 (96.1, 99.4) | 56.1 (52.3, 61.1) | 84.8 (84.6, 85.2) | 95.7 (87.3, 100.0) |
| NEUMISS | 76.4 (75.0, 78.2) | 67.0 (64.9, 70.1) | 79.0 (78.4, 79.5) | 75.4 (74.9, 75.9) | 98.6 (97.3, 99.4) | 84.1 (83.0, 85.2) | 85.5 (85.2, 85.7) | 100.0 (100.0, 100.0) |
| MINTY | 70.6 (67.4, 73.8) | 1.3 (0.3, 2.5) | 76.6 (75.2, 77.9) | 22.8 (8.0, 43.6) | 96.2 (94.1, 98.3) | 21.9 (16.8, 27.2) | 81.9 (80.1, 83.8) | 0.3 (0.0, 0.8) |
| MA-LASSO | 67.4 (65.7, 68.8) | 0.4 (0.1, 1.0) | 75.8 (74.9, 76.6) | 5.7 (5.4, 6.0) | 98.0 (96.6, 99.0) | 19.5 (19.0, 20.0) | 84.5 (84.0, 85.0) | 0.4 (0.0, 0.9) |
| MA-DT | 74.0 (71.8, 75.9) | 0.3 (0.0, 0.7) | 73.7 (73.0, 74.5) | 5.7 (5.4, 6.0) | 91.2 (86.9, 94.7) | 10.5 (10.2, 10.9) | 82.4 (81.6, 83.2) | 0.0 (0.0, 0.0) |
| MA-RF | 78.0 (76.2, 80.3) | 3.0 (0.6, 7.5) | 76.0 (75.4, 76.4) | 5.8 (5.5, 6.1) | 97.5 (96.7, 98.1) | 24.6 (20.9, 28.2) | 84.0 (83.3, 84.6) | 0.2 (0.0, 0.6) |
| MA-GBT | 77.0 (74.1, 79.9) | 3.1 (0.9, 6.6) | 75.7 (74.5, 77.2) | 6.1 (5.6, 6.6) | 96.7 (94.0, 98.6) | 18.0 (13.8, 21.4) | 83.1 (82.1, 84.1) | 0.4 (0.0, 1.1) |

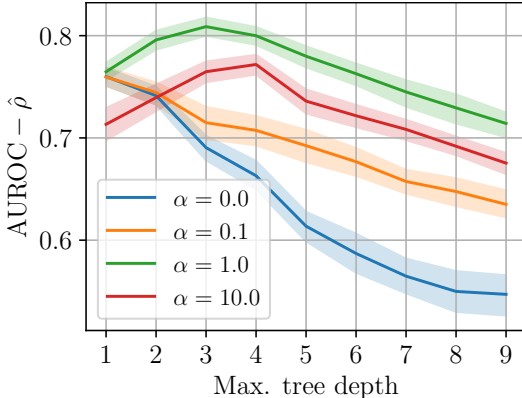

*Figure 5.* LIFE: AUROC $- \hat{\rho}$ vs. max. tree depth.

of the missingness reliance parameter $\alpha$. Figure 5 illustrates the trade-off between AUROC and missingness reliance ($\hat{\rho}$) when varying the maximum allowed tree depth in LIFE, across different $\alpha$ values. Finally, Figure 6 shows example LIFE trees under different $\alpha$ settings.

*Table 5.* Test-set AUROC and missingness reliance ($\hat{\rho}$) across two datasets, Breast Cancer and Pharyngitis, using zero ($I_M$) and MICE ($I_M$) imputation. The 95 % confidence intervals are based on the bootstrap distribution of the respective metric. For each dataset and MA estimator, $\alpha$ is selected as part of the overall model selection strategy, with the model minimizing $\hat{\rho}$ chosen among those achieving at least 95 % of the maximum AUROC.

| ESTIMATOR | BREAST CANCER | | PHARYNGITIS | |
|---|---|---|---|---|
| | AUROC ($\uparrow$) | $\hat{\rho}$ ($\downarrow$) | AUROC ($\uparrow$) | $\hat{\rho}$ ($\downarrow$) |
| LR ($I_0$) | 75.0 (73.4, 76.6) | 29.5 (17.8, 41.3) | 70.5 (68.1, 72.4) | 15.9 (12.8, 20.0) |
| LR ($I_M$) | 75.8 (74.2, 77.4) | 40.2 (39.0, 41.9) | 70.5 (68.2, 72.4) | 15.9 (12.8, 20.0) |
| DT ($I_0$) | 70.6 (68.5, 72.8) | 0.0 (0.0, 0.0) | 65.5 (63.0, 67.8) | 7.9 (4.7, 10.9) |
| DT ($I_M$) | 70.0 (67.5, 72.5) | 0.0 (0.0, 0.0) | 65.6 (63.0, 68.1) | 7.8 (4.6, 10.9) |
| RF ($I_0$) | 74.3 (72.1, 76.3) | 42.6 (41.8, 43.8) | 73.7 (72.1, 75.0) | 25.0 (22.5, 27.2) |
| RF ($I_M$) | 74.5 (72.5, 76.4) | 42.6 (41.8, 43.8) | 73.6 (72.2, 74.7) | 25.3 (22.6, 27.8) |
| M-GAM (IND) | 71.7 (69.3, 74.0) | 8.6 (0.3, 24.4) | 71.7 (70.2, 72.9) | 15.6 (12.9, 19.0) |
| M-GAM (INT) | 72.5 (69.4, 75.7) | 9.7 (0.5, 26.1) | 71.3 (70.2, 71.9) | 13.7 (12.8, 15.0) |
| XGBOOST | 74.7 (73.1, 76.5) | 42.4 (41.6, 43.6) | 72.4 (69.8, 75.3) | 19.9 (15.9, 24.1) |
| NEUMISS | 75.5 (73.5, 77.4) | 42.6 (41.8, 43.8) | 73.4 (71.2, 75.5) | 25.7 (22.8, 28.2) |
| MINTY | 70.9 (68.8, 73.9) | 32.6 (31.1, 34.5) | 69.7 (68.2, 70.9) | 11.8 (9.1, 13.8) |
| MA-LASSO ($I_0$) | 71.6 (69.7, 74.2) | 0.0 (0.0, 0.0) | 70.2 (67.7, 72.1) | 18.8 (16.6, 20.4) |
| MA-LASSO ($I_M$) | 71.0 (68.5, 74.1) | 8.4 (0.0, 25.2) | 70.2 (67.7, 72.1) | 18.8 (16.6, 20.4) |
| MA-DT ($I_0$) | 70.6 (68.6, 72.6) | 0.0 (0.0, 0.0) | 66.6 (63.2, 69.4) | 4.3 (2.4, 6.2) |
| MA-DT ($I_M$) | 70.6 (68.6, 72.6) | 0.0 (0.0, 0.0) | 66.9 (63.2, 69.6) | 4.3 (2.4, 6.2) |
| MA-RF ($I_0$) | 72.2 (69.7, 74.7) | 0.6 (0.1, 1.5) | 71.7 (69.9, 74.0) | 14.1 (12.6, 15.7) |
| MA-RF ($I_M$) | 72.2 (69.7, 74.7) | 0.6 (0.1, 1.5) | 71.4 (69.5, 73.9) | 14.1 (12.6, 15.7) |
| MA-GBT ($I_0$) | 70.8 (69.1, 72.1) | 0.6 (0.0, 1.2) | 68.2 (65.8, 71.0) | 6.8 (4.7, 8.8) |
| MA-GBT ($I_M$) | 71.2 (69.3, 72.7) | 0.7 (0.0, 1.5) | 68.2 (65.8, 71.0) | 6.8 (4.7, 8.8) |

*Table 6.* Test-set AUROC and missingness reliance ($\hat{\rho}$) across four datasets using zero imputation. The 95 % confidence intervals are based on the bootstrap distribution of the respective metric. For each dataset and MA estimator, we consider three different values for the missingness regularization parameter $\alpha$, where $\alpha = \infty$ and $\alpha = \alpha^*$ are selected as part of the overall model selection strategy. For $\alpha = \infty$, we select an $\alpha$ that is large enough to result in near-zero missingness reliance ($\hat{\rho} \approx 0$) while maximizing AUROC. For $\alpha = \alpha^*$, we select the model minimizing $\hat{\rho}$ chosen among those achieving at least 95 % of the maximum AUROC. Finally, $\alpha = 0$ effectively means no missingness regularization.

| ESTIMATOR | ADNI | | FICO | | LIFE | | NHANES | |
|---|---|---|---|---|---|---|---|---|
| | AUROC ($\uparrow$) | $\hat{\rho}$ ($\downarrow$) | AUROC ($\uparrow$) | $\hat{\rho}$ ($\downarrow$) | AUROC ($\uparrow$) | $\hat{\rho}$ ($\downarrow$) | AUROC ($\uparrow$) | $\hat{\rho}$ ($\downarrow$) |
| MA-LASSO ($\alpha = \infty$) | 50.3 (49.1, 51.5) | 0.7 (0.3, 1.2) | 50.0 (50.0, 50.0) | 0.0 (0.0, 0.0) | 57.1 (50.0, 71.3) | 0.1 (0.0, 0.2) | 63.7 (50.0, 77.4) | 0.2 (0.0, 0.7) |
| MA-LASSO ($\alpha = \alpha^*$) | 68.4 (66.1, 70.6) | 11.8 (0.1, 34.7) | 75.8 (74.9, 76.6) | 5.7 (5.4, 6.0) | 97.7 (96.7, 98.6) | 21.0 (19.1, 24.1) | 84.5 (84.0, 85.0) | 0.4 (0.0, 0.9) |
| MA-LASSO ($\alpha = 0$) | 69.0 (66.3, 72.1) | 66.5 (64.7, 68.6) | 76.4 (75.6, 77.4) | 75.4 (74.9, 75.9) | 98.7 (97.7, 99.4) | 84.1 (83.0, 85.2) | 77.5 (76.7, 78.2) | 100.0 (100.0, 100.0) |
| MA-DT ($\alpha = \infty$) | 70.2 (65.3, 75.3) | 0.1 (0.0, 0.3) | 75.7 (74.9, 76.4) | 5.7 (5.4, 6.0) | 71.8 (67.1, 76.8) | 0.2 (0.0, 0.3) | 83.9 (83.5, 84.3) | 0.0 (0.0, 0.0) |
| MA-DT ($\alpha = \alpha^*$) | 74.1 (71.7, 76.4) | 0.1 (0.0, 0.3) | 73.8 (73.0, 74.5) | 5.7 (5.4, 6.0) | 89.7 (87.5, 91.8) | 12.3 (10.4, 15.3) | 82.4 (81.6, 83.2) | 0.0 (0.0, 0.0) |
| MA-DT ($\alpha = 0$) | 72.7 (71.4, 74.2) | 11.7 (6.3, 18.8) | 73.8 (73.0, 74.5) | 5.7 (5.4, 6.0) | 92.2 (90.1, 93.6) | 18.3 (15.8, 19.9) | 82.3 (81.5, 83.0) | 0.1 (0.0, 0.2) |
| MA-RF ($\alpha = \infty$) | 71.7 (68.6, 75.8) | 0.9 (0.3, 1.5) | 75.3 (73.3, 76.7) | 5.7 (5.4, 6.0) | 75.5 (67.1, 87.9) | 5.7 (0.0, 16.8) | 82.5 (80.4, 84.2) | 0.0 (0.0, 0.1) |
| MA-RF ($\alpha = \alpha^*$) | 78.0 (76.2, 80.4) | 3.2 (0.7, 7.6) | 76.3 (75.6, 76.8) | 5.8 (5.5, 6.1) | 97.2 (95.7, 98.3) | 24.2 (22.0, 26.5) | 84.0 (83.3, 84.6) | 0.2 (0.0, 0.5) |
| MA-RF ($\alpha = 0$) | 78.4 (76.6, 80.5) | 57.5 (52.1, 62.9) | 75.7 (74.8, 76.5) | 12.2 (8.6, 15.8) | 95.2 (93.6, 96.4) | 20.8 (18.9, 23.8) | 84.2 (83.5, 84.9) | 0.4 (0.2, 0.6) |
| MA-GBT ($\alpha = \infty$) | 68.5 (64.7, 72.3) | 0.4 (0.1, 0.6) | 73.0 (71.4, 74.6) | 5.7 (5.4, 6.0) | 93.5 (91.1, 96.1) | 13.8 (9.7, 18.4) | 81.1 (79.2, 82.7) | 0.0 (0.0, 0.0) |
| MA-GBT ($\alpha = \alpha^*$) | 78.5 (76.4, 80.5) | 1.6 (0.7, 2.7) | 75.6 (74.5, 77.1) | 5.9 (5.5, 6.2) | 95.9 (92.1, 98.9) | 16.5 (11.9, 21.2) | 83.2 (82.1, 84.4) | 0.3 (0.0, 1.0) |
| MA-GBT ($\alpha = 0$) | 73.8 (71.8, 76.1) | 26.6 (10.9, 43.4) | 74.8 (73.8, 76.0) | 7.6 (5.4, 11.3) | 92.6 (90.3, 94.3) | 19.4 (18.8, 19.9) | 82.7 (81.9, 83.4) | 0.0 (0.0, 0.1) |

*Table 7.* Test-set AUROC and missingness reliance ($\hat{\rho}$) across two datasets, Breast Cancer and Pharyngitis, using zero imputation. The 95 % confidence intervals are based on the bootstrap distribution of the respective metric. For each dataset and MA estimator, we consider three different values for the missingness regularization parameter $\alpha$, where $\alpha = \infty$ and $\alpha = \alpha^*$ are selected as part of the overall model selection strategy. For $\alpha = \infty$, we select an $\alpha$ that is large enough to result in near-zero missingness reliance ($\hat{\rho} \approx 0$) while maximizing AUROC. For $\alpha = \alpha^*$, we select the model minimizing $\hat{\rho}$ chosen among those achieving at least 95 % of the maximum AUROC. Finally, $\alpha = 0$ effectively means no missingness regularization.

| | **BREAST CANCER** | | **PHARYNGITIS** | |
| --- | --- | --- | --- | --- |
| ESTIMATOR | AUROC ($\uparrow$) | $\hat{\rho}$ ($\downarrow$) | AUROC ($\uparrow$) | $\hat{\rho}$ ($\downarrow$) |
| MA-LASSO ($\alpha = \infty$) | 71.9 (70.3, 74.3) | 0.0 (0.0, 0.0) | 54.9 (51.5, 58.2) | 0.0 (0.0, 0.0) |
| MA-LASSO ($\alpha = \alpha^*$) | 71.6 (69.7, 74.2) | 0.0 (0.0, 0.0) | 70.2 (67.7, 72.1) | 18.8 (16.6, 20.4) |
| MA-LASSO ($\alpha = 0$) | 76.0 (74.5, 77.5) | 42.6 (41.8, 43.8) | 70.4 (68.2, 73.1) | 25.7 (22.8, 28.2) |
| MA-DT ($\alpha = \infty$) | 70.6 (68.6, 72.6) | 0.0 (0.0, 0.0) | 54.6 (50.8, 58.9) | 0.0 (0.0, 0.0) |
| MA-DT ($\alpha = \alpha^*$) | 70.6 (68.6, 72.6) | 0.0 (0.0, 0.0) | 66.6 (63.2, 69.4) | 4.3 (2.4, 6.2) |
| MA-DT ($\alpha = 0$) | 70.6 (68.6, 72.6) | 0.0 (0.0, 0.0) | 65.6 (63.0, 68.1) | 7.8 (4.6, 10.9) |
| MA-RF ($\alpha = \infty$) | 69.6 (67.5, 71.8) | 0.1 (0.0, 0.3) | 60.2 (55.3, 65.0) | 1.6 (0.3, 3.1) |
| MA-RF ($\alpha = \alpha^*$) | 72.2 (69.7, 74.7) | 0.6 (0.1, 1.5) | 71.7 (69.9, 74.0) | 14.1 (12.6, 15.7) |
| MA-RF ($\alpha = 0$) | 73.9 (72.7, 75.1) | 6.4 (2.6, 10.1) | 69.0 (66.0, 71.2) | 12.1 (9.6, 14.1) |
| MA-GBT ($\alpha = \infty$) | 69.6 (67.5, 71.6) | 0.0 (0.0, 0.0) | 57.6 (51.8, 65.2) | 1.2 (0.0, 2.6) |
| MA-GBT ($\alpha = \alpha^*$) | 70.8 (69.1, 72.1) | 0.6 (0.0, 1.2) | 68.2 (65.8, 71.0) | 6.8 (4.7, 8.8) |
| MA-GBT ($\alpha = 0$) | 71.4 (69.2, 73.3) | 1.0 (0.0, 2.1) | 68.6 (66.5, 70.5) | 9.3 (7.8, 11.2) |

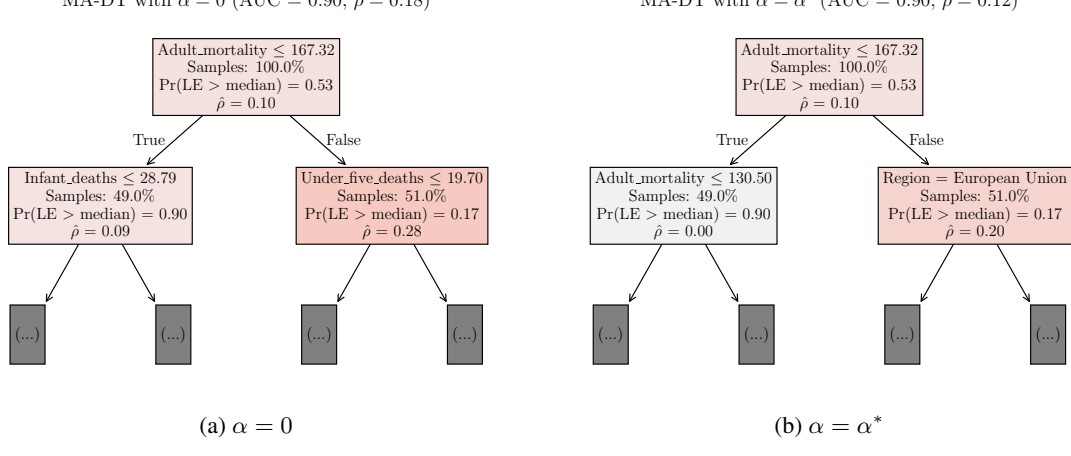

(a) $\alpha = 0$             (b) $\alpha = \alpha^*$

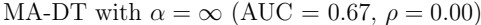

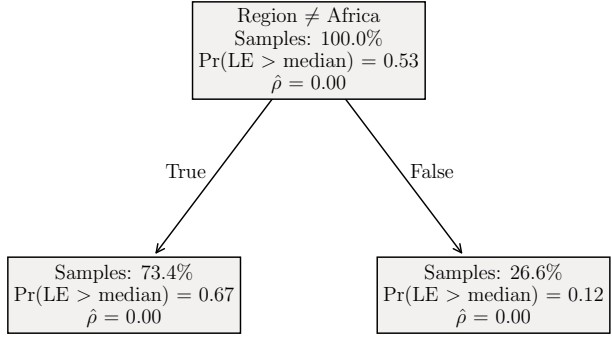

(c) $\alpha = \infty$

*Figure 6.* Example decision trees for $\alpha = 0$, $\alpha = \alpha^*$, and $\alpha = \infty$ fit to LIFE. The nodes are colored based on the missingness reliance $\rho$. The goal is to predict whether a country's life expectancy (LE) is above or below the median life expectancy. (a): `MA-DT` with $\alpha = 0$ behaves as a regular decision tree, splitting on highly predictive features such as `Adult_mortality` (probability of dying between 15 and 60 years per 1000 population) and `Infant_deaths` (number of infant deaths per 1000 population) (b): `MA-DT` with $\alpha$ selected to balance the trade-off between predictive performance and AUROC. Similar to the standard decision tree, `MA-DT` first splits on `Adult_mortality` at the root node, but then splits by the region corresponding to the European Union, reducing the missingness reliance for the overall tree. (c): With a large $\alpha$, the tree is built with no missingness reliance, only splitting by the region "Africa". While this tree achieves zero missingness reliance, it performs significantly worse than the others in terms of predictive accuracy.

