# OpenReview forum: "Prediction models that learn to avoid missing values"
_ICML.cc/2025/Conference — ICML 2025 spotlightposter_

### Official Review · Reviewer_eoMS · 2025-02-15

**Overall Recommendation:** 4

**Summary:**

The authors introduce missingness-avoiding (MA) machine learning, a framework for altering model training to avoid reliance on missing features. Through experiments on decision trees, lasso, and tree ensembles, they show that MA can reduce reliance on missing-value features with only a minor hit to predictive performance. This may yield improved interpretability of the model.

**Claims And Evidence:**

Yes, the experiments clearly show that MA estimators can mostly maintain performance while substantially reducing reliance on missing features.

**Essential References Not Discussed:**

None that I know of

**Experimental Designs Or Analyses:**

Yes, the experimental designs seem sound.

One arbitrary choice seems to be the selection of alpha, which is chosen as the candidate model with the lowest ρ among those achieving at least 95 percent of the maximum AUROC. This seems sensible but I am curious if there are other reasonable choices, particularly if any of them can lead to improved test AUROC (regardless of missingness reliance).

**Methods And Evaluation Criteria:**

Yes the proposed methods are clearly explained and the evaluation criteria make sense.

**Other Comments Or Suggestions:**

N/a

**Other Strengths And Weaknesses:**

The paper addresses an interesting problem and a clear and well-explained solution. Some of the novelty may be slightly limited in light of Stempfle & Johansson 2024, but the extension to trees seems to be quite useful in practice.

**Questions For Authors:**

Can the authors show, even qualitatively, examples where the decrease in missingness reliance translates to real-world importance, e.g. improved interpretability on one of the datasets?

**Relation To Broader Scientific Literature:**

The results generalize the results of Stempfle & Johansson 2024, which are specific to generalized linear rule models

**Theoretical Claims:**

N/A

---

> ### Author Rebuttal · Authors · 2025-03-31
>
> We thank the reviewer for their valuable feedback and appreciate their recognition of the clear explanations and sensible evaluation criteria. We respond to specific comments below.
>
> **Re: real-world importance of reducing missingness reliance**
>
> Thanks for raising this point—we understand its importance in demonstrating the improved interpretability of our MA method. In the last paragraph of Section 6.2, we refer to MA trees fitted to the Life dataset, shown in Figure 6 in the appendix. This example demonstrates how MA-DT behaves under different values of the missingness regularization parameter $\alpha$. For instance, when $\alpha$ is tuned to balance accuracy and reliance on missing values, MA-DT reduces reliance on missing data by 33% without sacrificing accuracy compared to a standard decision tree. We will consider including these figures in the main text in a potential camera-ready version. Additionally, we will consider adding examples of MA trees fitted to the ADNI dataset. These examples illustrate how MA learning avoids splitting on features that are likely to be missing, instead favoring more complete features that maintain predictive performance. At test time, such trees are more interpretable since they do not rely on features with missing values.
>
> **Re: different strategies for choosing $\alpha$**
>
> In the submitted version of the work, we primarily considered two strategies for selecting $\alpha$. We chose $\alpha=\alpha^{*}$ as part of the overall model selection process, selecting the candidate model with the lowest missingness reliance among those achieving at least 95% of the maximum AUC. If predictive performance is a priority, this percentage could be increased (at the potential cost of increased missingness reliance). Conversely, we chose $\alpha=\infty$ by selecting the model with the highest AUC among those achieving near-zero missingness reliance (ensuring that the range of $\alpha$ included sufficiently high values). A more general approach could involve setting a threshold for missingness reliance, such as 20%, and then maximizing AUC among models meeting this threshold. Another alternative is to maximize $\mathrm{AUC} – \gamma\cdot\hat{\rho}$, where $\gamma$ controls the trade-off between AUC and missingness reliance and needs to be set for the specific problem at hand.
>
> We thank the reviewer for raising this point and we have added the following discussion to the revised paper: Alternative strategies for selecting $\alpha^*$ should consider the specific application needs. For instance, rather than focusing solely on the acceptable trade-off in predictive performance for achieving the lowest possible $\alpha$, we might prioritize limiting the number of missing features per individual. Relying on an average measure across the dataset may not provide sufficient insight. Additionally, domain knowledge can help define an acceptable level of missingness, but quantifying this threshold is often challenging and subjective.

---

### Official Review · Reviewer_HDgG · 2025-03-01

**Overall Recommendation:** 3

**Summary:**

This manuscript focuses on improving the reliability of machine learning methods when encounter missing value at inference time. A novel framework termed Missingness-Avoiding learning is proposed to reduce the models' reliance of missing values for decision trees, sparse linear models and ensemble methods. Specifically, a classifier-specific regularization technique is proposed to alivate the models' dependency on missing values. Experiments on six datasets are carried out to validate the proposed method.

**Claims And Evidence:**

First of all, I would like to acknowledge that I am not familiar with the related works. I tried my best to understand this work and do the review. It is highly possible that I miss some importance aspects of this submission.

- The authors claim that the proposed method is a general framework for training models with missing values. However, the proposed MA is only implemented with three types of classic machine learning methods, i.e., decision tree, linear model and ensembles. Is MA applicable for more general models like neural networks?

**Essential References Not Discussed:**

None.

**Experimental Designs Or Analyses:**

- The experimental designs make sense to me. I have no further comments on this point.

**Methods And Evaluation Criteria:**

- Most of the datasets used in the experiments are small-scale tabular data, which may be insufficient to well support the proposed method.

- The compared baselines may be insufficient. For examople, the proposed MA is only compared with zero-imputation and MICE. Other advanced imputation-based methods such as [1] is missed.

[1] GAIN: Missing Data Imputation using Generative Adversarial Nets, NIPS'18

[2] TabDDPM: Modelling Tabular Data with Diffusion Models, ICML'23

**Other Comments Or Suggestions:**

None.

**Other Strengths And Weaknesses:**

None.

**Questions For Authors:**

None.

**Relation To Broader Scientific Literature:**

None.

**Theoretical Claims:**

There is no proofs in this manuscript that need to be checked.

---

> ### Author Rebuttal · Authors · 2025-03-31
>
> We thank the reviewer for their insightful feedback and appreciate their recognition of our work addressing a practically important yet underexplored challenge in the healthcare domain—enhancing model reliability under test-time missingness. We address the specific comments below.
>
> **Re: MA learning in neural networks**
>
> Standard feed-forward neural networks typically rely on all input features to a non-zero degree unless the learning objective is regularized to yield sparse weights. Even then, the reliance is generally not contextual but remains the same for every input, regardless of the observed features. In principle, contextual reliance could be achieved by combining the MA penalty with attention modules that attend only to observed features. However, our focus is on tabular data, where neural networks tend to underperform. We instead target model classes like decision trees and linear models, which not only perform well on tabular data but also offer interpretability—an essential requirement in our primary application domain, healthcare. Exploring MA learning with attention modules for tabular representations is an interesting direction for future work.
>
> **Re: dataset sizes**
>
> We thank the reviewer for their observation and agree that the number of features in our datasets is relatively small (ranging from 16 to 42). However, we include a diverse set of datasets in terms of sample size—from a few hundred (e.g., Pharyngitis: 676, ADNI: 1,337, Breast Cancer 1,756, LIFE 2,864) to over 10,000 samples (e.g., FICO: 10,549, NHANES: 10,000). These dataset sizes are typical in the tabular ML literature and reflect realistic constraints in many application domains.
>
>
> **Re: advanced imputation methods as baselines**
>
> Thank you for sharing the two works. We agree that the methods could be compared to more advanced imputation approaches. However, our goal for the baselines was to include representative methods from different strategies for handling missing data: impute-then-predict, model-specific handling of missingness, and treating missingness as an informative signal. Each of these serves a slightly different purpose compared to the MA approach. We chose zero imputation and MICE because they are both widely studied in the literature and commonly used in practice, making them strong reference points for comparison. More advanced imputation methods are likely to yield similar reliance on missing values when combined with similar regressions or classifiers. Thus, the choice of imputation method is secondary in our study. To enable a fair comparison with our method, we also calculate the missingness reliance for the baselines. Since the baseline methods were not designed with the goal of learning to avoid missing values, but rather to provide accurate imputations, they may be incentivized to rely on well-imputed values—potentially leading to a higher missingness reliance metric.

---

### Official Review · Reviewer_go58 · 2025-03-06

**Overall Recommendation:** 4

**Summary:**

This work introduces a generic framework for encouraging models to avoid accessing missing values through regularization. Specific implementations of this framework for Lasso, greedy decision trees, and tree ensembles are introduced. A thorough discussion of the settings in which missingness can and cannot safely be avoided is included, followed by experiments showing that the level of reliance on missing features can be substantially reduced without sacrificing much predictive performance on several real world datasets.

**Claims And Evidence:**

Yes. The clear discussion of settings in which the MA framework is expected to fail (Section 5.2) is particularly appreciated, and helps clarify when MA can safely be used.

**Essential References Not Discussed:**

(w2) One notable omission from the set of baselines considered is [1]. Similar to this work, [1] introduces an approach that aims to avoid including missing values in the splits used in a decision tree. This is particularly relevant because [1] aims to split on x_j only in the subspace where x_j is not missing; this is an approach that would perform particularly well in the ODDC rule setting considered in the present work.

[1] Beaulac, Cédric, and Jeffrey S. Rosenthal. "BEST: A decision tree algorithm that handles missing values." Computational Statistics 35.3 (2020): 1001-1026.

**Experimental Designs Or Analyses:**

I did not check the code, but the description of the experimental setup is clear and thorough, and the described experimental setup makes sense.

**Methods And Evaluation Criteria:**

Yes.

**Other Comments Or Suggestions:**

To help respond to this review, I've labelled each point on which a response would be appreciated with (w#).

Some additional suggestions to which I do not expect a response:
- It is somewhat confusing to introduce sigma_ij in section 4.1 — I would recommend holding off on adding this term until 4.3, and simply defining an updated splitting rule there.
- 4.2 describes an l1 based regularization, but the target problem seems to suggest an L0 regularization (i.e., we want to guide coefficients to exactly 0). It may be interesting to consider methods that optimize L0 regularized classification directly (e.g., [2, 3]) rather than Lasso. I do not expect this experiment to be done for rebuttal, and it does not effect my rating.
- Throughout the paper there is discussion using the language of regression (i.e., “impute-then-regress”). However, the experiments focus on classification. Unifying the language would improve readability.
- Typos
    - Line 162, right column — “we propose medications” should be “we propose modifications”
    - Line 216 left — it would be more accurate given equation 5 to say that Lasso uses a parameter lambda > 0, rather than alpha > 0
    - Line 194 right — “individual trees ar fit” should be “individual trees are fit”

[2] Liu, Jiachang, et al. "Fast sparse classification for generalized linear and additive models." Proceedings of machine learning research 151 (2022): 9304.
[3] Dedieu, Antoine, Hussein Hazimeh, and Rahul Mazumder. "Learning sparse classifiers: Continuous and mixed integer optimization perspectives." Journal of Machine Learning Research 22.135 (2021): 1-47.

**Other Strengths And Weaknesses:**

Strengths
- The paper is quite well written, with clean and clear notation used throughout.
- The proposed methods are elegant, and empirically shown to be effective.
- This work provides a clear discussion of settings in which we expect MA regularization to be viable in general (5.1) and in which we don't expect MA regularization to be viable (5.2). This is very helpful in understanding the method, and will support future research.

Weaknesses
- (w1) See "Theoretical Claims"
- (w2) See "Essential References Not Discussed"
- (w3) An (arguably) less restrictive setting than MNAR in which this strategy may be expected to fail is the case of informative missingness. This setting is discussed in Appendix C, but is not connected with the main paper in any way. I recommend adding some discussion of this point to the end of 5.2, since, in this case, we would expect MA to find inferior accuracy to missingness indicator approaches.
- (w4) See "Questions For Authors" <- This is my primary concern, and I am inclined to increase my score if it can be addressed in a compelling way.
- (w5) See "Questions For Authors"

**Questions For Authors:**

- (w4) Why should practitioners prefer to avoid accessing missing variables, rather than reasoning based on missingness/handling it natively with, e.g., defaults? This seems like a key question in motivating this work, but is not compellingly answered in the current version of the paper. This is my primary concern, and I am inclined to increase my score if it can be addressed in a compelling way.
- (w5) Corollary 1 speaks to the existence of a Baye's optimal model with 0 reliance on missing features, but it says nothing about convergence/relevance of the given loss functions to finding this h*. Is it possible to say anything stronger/connect the optimization problems described in Section 4 to this Corollary?

**Relation To Broader Scientific Literature:**

This work takes a different angle on the well-studied problem of handling missing data. It appropriately considers impute-then-predict, model-specific missingness handling, and missingness as a value baselines, which each have slightly different goals than the MA approach.

**Theoretical Claims:**

Yes; I read all three proofs in Appendix B. The proofs are correct, although I have a minor suggestion for Corollary 1. (w1) The given proof is valid, but does not specify the hypothesis class from which h* is drawn. From the notation used in the paper, it is implied that h* must be a member of the hypothesis class H we are currently considering. This is benign for universal approximators like decision trees, but may be misleading when H is the class of linear models (there may be no linear model h st h(x) = E[Y | x] ). For clarity, I recommend specifying that h* may not be a member of H where appropriate.

---

> ### Author Rebuttal · Authors · 2025-03-31
>
> We thank the reviewer for their thoughtful and valuable feedback, including their appreciation of our methods, empirical effectiveness, and the helpful discussion. We have addressed all suggestions to improve clarity and respond to specific comments below.
>
> **Re: motivation of avoiding missing values (w4)**
>
> MA learning is designed for prediction tasks with missing data where interpretability matters—clinical risk scores are a key example. Standard approaches (imputation, missingness indicators, default rules) often yield models that rely on unavailable inputs, as shown by their high missingness reliance in our experiments, which can reduce interpretability. For example, it may be difficult to justify that a physician or model should guess the value of a missing test (if using imputation) or that a patient’s risk of readmission should go down because the test result is missing (if using indicators or default rules in trees)—especially if the test is irrelevant based on the values of other features, such as a diagnosis. Though such tests may be predictive in other contexts, many learning tasks are underspecified, and models with similar accuracy may differ in missingness reliance. MA learning favors models that use information that is both predictive *and* available. Finally, we argue that MA learning and reasoning about missingness patterns go hand in hand. Increasing the penalty $\alpha$ encourages trees to split first on commonly observed features (e.g., demographics), and then on contextually available ones (e.g., tests for certain conditions). As discussed in Section 5 on ODDC rules, these observation patterns may be correlated with predictiveness, e.g., when the outcome reflects a diagnosis based on observed information.
>
> **Re: relevant literature (w2)**
>
> We thank the reviewer for highlighting the work by Beaulac & Rosenthal and have added it to the related work section. As noted, BEST suits ODDC settings with clear knowledge of the data-generating process, but defining gating variables is less straightforward with limited knowledge. In this sense, we view the MA framework as more general. A key difference between our methods is that MA trees do not explicitly split on the missingness mask. On the one hand, BEST can explicitly build trees using mask splits—often at the top—approaching the pattern submodel (PSM) of Mercaldo & Blume. On the other hand, MA trees cannot split based on non-MAR patterns, which may limit their expressiveness in such cases. However, avoiding explicit mask splits improves interpretability by preventing the tree structure from being dominated by missingness logic. Comparing MA-DT to BEST on the dataset from Beaulac and Rosenthal would be interesting, but the dataset does not appear to be publicly available.
>
> **Re: informative missingness (w3)**
>
> We introduce informative missingness in Section 3 but agree that Appendix C felt somewhat isolated. The reviewer is correct: when missingness is informative, adding missingness indicators is generally preferable to using an MA approach. For example, Beaulac & Rosenthal show that when missingness depends on the label, indicators can be predictive. In such cases, MA trees cannot leverage this information and may underperform, depending on the imputation. We appreciate the feedback and address this in the revised manuscript (Section 5.2).
>
> **Re: convergence (w5)**
>
> The reviewer raises an important question: if an optimal model $h^*$ exists, can our algorithm recover it with enough data? This depends on the model class, optimization, and data distribution. For example, if $h^*$ is a sparse logistic model, our L1 objective can likely recover it—similar to LASSO. The optimization in Section 4 supports Corollary 1: under ODDC rules, we can learn models with low missingness reliance and strong performance. We will add this discussion to the final paper.
>
> **Re: clarity (w1), notation, and regularization**
>
> - We agree it's important to clarify the relationship between $h^*$ and $\mathcal{H}$ and now state explicitly that $h^*(x) = \mathbb{E}[Y \mid x]$ may lie outside $\mathcal{H}$. This is benign for universal approximators (e.g., decision trees) but more significant for restricted classes like linear models.
> - We agree that introducing $\sigma_{i,j}$ in Section 4.1 may be premature; we now assume equal feature contribution initially and introduce $\sigma_{i,j} \in {0,1}$ later in Section 4.3, where it supports the generalization to tree ensembles.
> - We also acknowledge that starting with L0 regularization to penalize missing features would have been natural; however, we focused on L1 (LASSO) due to its compatibility with widely used implementations (e.g., scikit-learn, statsmodels). We will clarify this choice, along with a discussion of L0-based alternatives.
>
> Lastly, we thank the reviewer for their valuable feedback and excellent suggestions, and for noting inconsistencies and typos, which have been corrected in the revision.

---

> > ### Comment · Reviewer_go58 · 2025-04-01
> >
> > I thank the authors for their comprehensive response, and I am generally quite satisfied with the points raised by the authors. In particular, I found the clinical risk score example compelling -- a test might be given only to patients that a clinician considers high risk, so it does not help practitioners to say "this patient is low risk because they haven't had this test". I hope the authors will include this, and all other points raised in the rebuttal, in the final manuscript. I enjoyed reading this paper, and hope it will be accepted. I have updated my score accordingly.

---

> > > ### Author Response · Authors · 2025-04-07
> > >
> > > Thank you for your thoughtful feedback and kind words. We’re glad the clinical risk score example resonated with you and will ensure all points from the rebuttal are reflected in the final manuscript. We truly appreciate your support and updated score!

---

### Official Review · Reviewer_dApi · 2025-03-17

**Overall Recommendation:** 4

**Summary:**

The paper proposes a novel framework – “missingness-avoiding” (MA) machine learning – designed to mitigate the impact of missing data during prediction. The core idea is to train models that inherently minimize reliance on features with missing values at test time. This is achieved by incorporating classifier-specific regularization terms into the learning objectives for decision trees, sparse linear models, and ensemble methods.  The authors demonstrate this approach on several datasets, showing improved performance compared to standard imputation techniques while maintaining interpretability

**Claims And Evidence:**

The central claim – that models can be trained to actively avoid using missing values – is clearly articulated in the abstract and introduction. However, the level of detail regarding when this avoidance is feasible and effective isn’t consistently clear throughout the paper. The evidence supporting this claim is primarily demonstrated through experimental results across multiple datasets. While the quantitative performance improvements are notable (especially with MA-DT), the paper could benefit from a more rigorous discussion of the conditions under which these gains are most pronounced.

**Essential References Not Discussed:**

Most essential references are in the paper

**Experimental Designs Or Analyses:**

Experimental Design: The experimental design is commendable, utilizing multiple datasets (NHANES, LIFE, ADNI, Breast Cancer, Pharyngitis) with varying characteristics to demonstrate the generalizability of the MA approach. The inclusion of zero imputation as a baseline comparison is also sensible.

Analysis: The analysis effectively presents the quantitative results, clearly showing the performance gains achieved by the MA models compared to standard methods. However, the paper could benefit from a more in-depth discussion of why these improvements occur. For example, are the MA models better at capturing non-linear relationships that are obscured when missingness is treated as simple imputation? A qualitative analysis of the decision trees generated by the MA-DT model would also be valuable. The paper could also benefit from a more thorough comparison to stacked models trained on complete data

**Methods And Evaluation Criteria:**

Methods: The authors’ method is well-defined, outlining the specific regularization terms applied to different model types (decision trees, sparse linear models, ensemble methods). The incorporation of classifier-specific regularization is a clever and theoretically sound approach. The use of MA-DT, MA-LASSO, MA-RF, and MA-GBT provides a good range of model implementations for comparison.

Evaluation Criteria: The evaluation primarily focuses on AUROC (Area Under the Receiver Operating Characteristic curve) as a measure of predictive performance and missingness reliance. The use of cross-validation to assess model stability is appropriate. However, the paper could benefit from exploring other relevant metrics beyond AUROC, such as precision, recall, or F1-score, particularly when considering the implications for clinical applications where false positives and negatives have different costs.

**Other Comments Or Suggestions:**

Please consider expanding on the theoretical framework, especially around when MA-learning is suitable.

**Other Strengths And Weaknesses:**

The paper presents an interesting and potentially valuable concept with promising experimental results. However, the lack of clear guidance on when this approach is most effective, coupled with a somewhat superficial theoretical justification, warrants a "weak accept" recommendation. Further work is needed to solidify the theoretical foundations, provide more robust comparisons against alternative methods (particularly stacked models), and offer practical guidelines for practitioners on how to best apply this framework.

**Questions For Authors:**

See above

**Relation To Broader Scientific Literature:**

Overall the paper has sufficient literature review

**Theoretical Claims:**

The theoretical underpinning – that models can learn to exploit missingness patterns – is rooted in ideas related to Bayesian inference and the concept of minimizing expected risk. The paper does a reasonable job of connecting this with the idea of avoiding reliance on features with high missingness rates, but it could be strengthened by explicitly stating how the regularization terms are mathematically derived from these principles.  For example, it might be useful to compare this approach to simpler stacked models and use this to situate where MA-learning could be useful.

---

> ### Author Rebuttal · Authors · 2025-03-31
>
> We thank the reviewer for their insightful feedback and appreciate their recognition of our well-motivated and novel MA framework. We have addressed all suggestions to improve clarity and respond to specific comments below.
>
> **Re: relevance of MA learning**
>
> We would like to direct the reviewer to Sec. 5, where we outline when MA learning is expected to perform well (Sec. 5.1) and when it may face challenges (Sec. 5.2)—a discussion also highlighted as a strength by Reviewer go58. MA learning is expected to work well when the missingness mask follows a clear structure,  such as when ODDC rules govern the data-generating process. When there is no structure in the missingness patterns related to observed features, such as in an MCAR or a fully MNAR setting, we expect it to be harder for MA learning to achieve low missingness reliance with maintained accuracy. As the reviewer noted, our experiments show that MA learning outperforms baselines across datasets while greatly reducing reliance on missing values. Finally, MA learning may benefit from being combined with missingness indicators when missingness itself is highly informative—the reliance penalty will just make sure to keep such features to a minimum.
>
> **Re: guidelines for practitioners**
>
> To apply the MA framework in practice, we recommend starting by selecting a model class that aligns with the interpretability requirements of the application. Linear models and decision trees offer greater transparency, while ensembles may yield better performance with reduced interpretability. Once the model type is chosen, the trade-off between predictive performance and reliance on missing features can be adjusted through the regularization parameter $\alpha$. A higher $\alpha$ encourages the model to avoid relying on features that are often missing at test time, while a lower $\alpha$ prioritizes performance. The appropriate choice of $\alpha$ depends on how much missingness reliance is acceptable in the given context. We describe different strategies in the response to Reviewer eoMs. To evaluate this, practitioners can compute the reliance score ($\hat{\rho}$), which quantifies how frequently a model depends on missing features. Importantly, MA learning requires no assumptions about the underlying missingness mechanism (although the achievable accuracy-reliance tradeoff will be affected by its structure), making it a practical and robust approach across a wide range of real-world settings.
>
> **Re: qualitative analysis**
>
> We agree that qualitative analysis of MA trees is valuable. In Section 6.2, we refer to Figure 6 (appendix), which shows how MA-DT behavior varies with different values of the missingness regularization parameter $\alpha$ on the Life dataset. For instance, when $\alpha$ is tuned to balance accuracy and reliance on missing values, MA-DT reduces reliance on missing data by 33% without sacrificing accuracy compared to a standard decision tree. For the camera-ready version, we will consider including these figures in the main text and adding MA tree examples from the ADNI dataset. These illustrate how MA learning favors complete features over those with missingness, enhancing interpretability without compromising accuracy at test time.
>
> **Re: theoretical foundations and stacked models**
>
> Thank you for the insightful comment. We agree that strengthening the theoretical foundations is important. As for stacked models, we're unsure how they would address the challenges in our setting. Stacking typically involves training a meta-model on the outputs of several base models (e.g., decision trees, logistic regression). The reviewer suggests that these may be trained using complete data, but this is not generally available without imputation, which is what we try to avoid in the first place. Alternatively, we could train base models on different data subsets based on their missingness pattern, but this poses the same problem—it seems to us that the stacking question is orthogonal to reducing missingness reliance, but could be an alternative to boosting. We’re very open to considering this direction and would greatly appreciate any additional insights the reviewer can share on their suggestion.
>
> **Re: clarification on MA learning**
>
> Many model classes have large Rashomon sets—sets of near-optimal models that have similar predictive performance but differ on other properties. By adding the missingness reliance penalty, we prioritize models within this set that rely less on missing values. This can exploit nonlinear patterns (as with MA trees, where reliance is contextual) or linear ones (as with variable selection in MA-LASSO). We will include this in our Section 5 discussion.
>
> **Re: performance metrics**
>
> We agree that multiple metrics are important to capture different aspects of performance. Due to space constraints, we focused on AUROC and missingness reliance in the main text and will consider including additional metrics (e.g., F1 score) in the appendix.

---

### Decision · Program_Chairs · 2025-05-01

**Decision:**

Accept (spotlight poster)

**Comment:**

This submission contributes a framework to facilitate machine-learning predictions in the presence of missing values by adding corresponding regularization. It led to interest and discussion with the reviewers, which appreciated the prediction framework and the empricial study. The work could be improved by demonstrating the framework beyond trees.